# Stage-dependent cerebrocerebellar communication during sensorimotor processing

**Vincenzo Romano** [1,4] ✉, **Matthijs van Driessche** [1,4], **Nathalie van Wingerden**[1], **Staf Bauer** [1], **Brendan Boeser**[1], **Jorge F. Mejias** [2] ✉ & **Chris I. De Zeeuw**[1,3] ✉

Cerebral cortex and cerebellum are essential for sensorimotor control, but the dynamics of their interactions remain unclear. Here, we investigated which pathways prevail during preparation and execution of spontaneous whisker movements in mice. During preparation, neuronal activity of primary motor (M1) and somatosensory (S1) cortex precede that of cerebellar crus regions, with a lead that is consistent with relaying a copy of motor commands. After movement onset, the phase of the signal inverts, indicating a dominant vector signaling from cerebellum to cerebrum. At this stage, Purkinje cell activity correlates more with S1 than M1, generating a prediction of sensory consequences during motor actions. A computational cerebello-cortical model could replicate the changes in dynamics and directionality. Optogenetic manipulations of pons and thalamus confirm the modeled predictions on stage-dependent dynamics. Together our data point towards a swap in direction of information flow between cerebrum and cerebellum when motor preparation switches to execution.

The brain regulates motor behavior by relaying signals between cerebral cortex and cerebellum in a bidirectional fashion[1–4]. The main descending pathway from cerebral cortex to the cerebellum is relayed via the pons[5], while the ascending pathway from cerebellum to cerebrum exerts its main effects via the thalamus[6,7]. Together, these descending and ascending pathways form one of the most prominent loops of the central nervous system. It is not only involved in complex motor tasks, such as discrimination and reward learning through active exploration with whisker, limb and/or tongue movements[2,3], but also in the coordination of innate and ethological motor behaviors, such as spontaneous whisking or licking[8,9]. Despite the extensive knowledge on the connectivity and behavioral relevance of this loop[10–12], the neural dynamics that optimize coordination of sensorimotor behavior are still unclear.

Theoretical studies posit that coordination of natural motion is fast and precise, because internal models generated in the cerebrocerebellar system facilitate not only the execution, but also the planning of movements[13–15]. During the planning stage the cerebellum may receive a copy of the upcoming motor commands from cerebral cortex and generate a predictive representation of the sensory consequences of the action that is planned[16]. When the action is being executed, the actual somatosensory feedback will be compared with the sensory consequences predicted by the internal model of the cerebellar system, and if they don't match, a corrective output will be relayed to the cerebral cortex so as to better align the next motor command signals with the desired movement and corresponding feedback[16–18].

Even though it is attractive to hypothesize about the cerebrocerebellar system operating in such a feedforward fashion with feedback control, the electrophysiological evidence for this theory is still limited. In particular, it remains to be shown that the increased activation of pyramidal cells in the primary motor cortex (M1) precedes that of the

[1]Dept. of Neuroscience, Erasmus MC, Rotterdam, The Netherlands. [2]Swammerdam Institute for Life Sciences, University of Amsterdam, Amsterdam, The Netherlands. [3]Netherlands Institute for Neuroscience, Royal Dutch Academy of Arts & Sciences, Amsterdam, The Netherlands. [4]These authors contributed equally: Vincenzo Romano, Matthijs van Driessche. ✉e-mail: v.romano@erasmusmc.nl; j.f.mejias@uva.nl; c.dezeeuw@erasmusmc.nl

changes in modulation of Purkinje cells in the cerebellar hemispheres during the planning of movements, that the main direction of signaling between cerebrum and cerebellum alters around the moment when the planning is transitioning into the execution of the movement, and that the cerebellum holds a sensory representation of the ongoing actions in that its output activity correlates more with the activity of the primary sensory cortex (S1) than that of M1 during a proper execution. Accordingly, if the cerebellar system needs to generate a corrective motor signal to better align the actual with the desired movement, its output should affect M1 more than S1.

To address these predictions and questions, we set out to record cerebrocerebellar activity during spontaneous whisker movements in mice. Whisker movements are highly relevant for mice from an ethological perspective and their representations are prominent in both cerebral cortex and cerebellum[19–24]. Moreover, anatomical and physiological experiments have revealed how the cerebrocerebellar loop is superimposed on the brainstem circuitry that directly drives the whisker movements[25–28]. We found that, before movement onset, neuronal activity of M1 and S1 correlated with Purkinje cell activity of the crus regions, with a phase difference consistent with the cerebellum receiving a copy of the cerebral commands for whisker movements. After movement onset the phase of the correlation inverted, suggesting a dominant vector signaling from cerebellum to cerebrum. A simple feedforward model of the whisker system, comprising both the descending and ascending tracts between cerebral cortex and cerebellum, could readily replicate our electrophysiological results during the different stages. Moreover, the model allowed us to make predictions about what would happen if the activity of the neurons of the intermediate hubs of the descending and ascending tracts, i.e., the pons and thalamus, are manipulated. These predictions were met when we altered the activity of these cells with optogenetics. Indeed, the changes in activity in the downstream areas involved were congruent with the changes predicted by the stage-dependent effects that we modeled. Together, our results indicate that the cerebellum integrates sensory and motor information from cerebral cortex in a stage-dependent manner to compute an output that can adjust motor output via premotor areas in the brainstem, while simultaneously providing feedback to neocortex.

## Results

### The phase of cerebrocerebellar communication flips from planning to movement

To investigate the temporal dynamics of cerebrocerebellar communication, we recorded local field potential (LFP) signals from the whisker motor cortex (M1) and barrel cortex (S1) as well as from the whisker regions in crus 1 and 2 of the lateral cerebellum during different stages of natural whisking behavior (Fig. 1a and Fig. S1; see Methods section). We focused our analysis on the 200 ms period before (Pre-movement) and the 200 ms period after (Movement) the onset of self-initiated whisker movements (Fig. 1b). During the Movement period mice could perform protractions, retractions or both. For all possible combinations of the 16 recording channels in M1, the 16 recording channels in S1, and the 12 recording channels in the Purkinje cell layer of the crus regions of the cerebellar cortex we calculated the phase relations (i.e., time displacement) of the cross-correlograms of the LFPs before and after onset of a protraction or retraction (for data of one exemplary mouse, see Fig. 1c, d and Fig. S1). At the population level ($n = 12$ mice), we found that the phase relation switched from a lead of $-2.36 \pm 0.46$ ms (i.e., cerebral signals leading those of the cerebellum) to a lag of $+2.51 \pm 0.57$ ms (i.e., cerebral signals lagging those of the cerebellum) for M1, once the movement started (Fig. 1e, f, left). Likewise, for S1 we found a similar lead ($-1.86 \pm 0.57$ ms) and lag ($+2.56 \pm 1.20$ ms) for the Pre-movement stage and Movement stage, respectively (Fig. 1f, right; for details across different cortical layers, see Fig. S2). Therefore, the phase relationship between cerebral and

cerebellar LFPs signals appears to switch from Pre-movement to Movement epochs for M1 ($p = 0.00005$; paired TTEST, see also Table 1 for details on the statistical tests) and to a lesser extent also for S1 ($p = 0.00512$; paired TTEST). These conclusions on the stage-dependent phase relationships of the LFP signals are well in line with the current flow of LFP signals from M1 to S1 and with that from the superficial to the deeper layers in M1 and S1 during whisker movements[29–31] (Fig. S3a−c).

Analyses of the spike data recorded in M1 and cerebellum before and after movement onset showed the same phase relationships as revealed by the LFP signals (Fig. S4), despite the fact that the spike data had lower mutual information than the LFP signals ($p \leq 0.001$; Mann−Whitney $U$-test; Fig. S4a, b). Both Granger causality analysis ($p = 0.0051$; paired TTEST; Figure S4c) and trial-by-trial variance analysis[23,24,32] of the spikes ($p = 0.023$; paired TTEST; Fig. S4d) revealed the same switch when comparing Pre-movement and Movement epochs. Yet, when considering the spike data recorded in S1, our analyses did not reach statistical significance ($p = 0.1686$ and $p = 0.1090$, respectively, paired TTEST; Fig. S4c,e). Thus, the spiking data confirm the LFP data in that the relationship of activity in M1 and the cerebellar crus regions switches from a cerebrocerebellar direction during the Pre-movement stage to a cerebellocerebral direction during the Movement stage.

### Purkinje cell activity correlates more with sensory than motor cortical activity

If the cerebellum holds a sensory representation of well-executed, ongoing actions, one can expect that activity in the cerebellar cortex correlates more with that of S1 than that of M1 and that this correlation is strongest in the execution stage. To test these hypotheses we calculated correlation strengths between LFPs recorded at the Purkinje cell layer level of the crus regions and LFPs of S1 and M1 during the Pre-movement and Movement periods (Fig. 2a). As predicted, in both of these 200 ms time windows, the cerebellar activity showed a stronger correlation with S1 than M1 activity (for Pre-movement $p = 0.032$, for Movement $p = 0.019$; paired TTESTs), (Fig. 2b). In addition, the correlation between S1 and cerebellum was stronger during the Movement period than during the Pre-movement period ($p < 0.001$; paired TTEST), (Fig. 2c). These findings held true for all the 12 different cerebellar locations in which Purkinje cell activity was recorded. Indeed, when plotting the differences between the S1 and M1 correlations across the schematic representation of the cerebellar hemisphere during the Movement period, we could not find any exception to the finding that cerebellar activity correlated more with S1 than M1 activity (Fig. 2d). Next we compared the modulation of the spiking activity of single-unit activity of Purkinje cells and single-unit activity of presumptive pyramidal cells in the cerebral cortex (Fig. 2e)[33] and we found that overall the magnitude of the correlation of Purkinje cell-S1 pairs was higher than that of Purkinje cell-M1 pairs (Fig. 2f top, $p = 0.0031$; unpaired TTEST). Accordingly, when we quantified the percentage of pairs in which the modulation of single-unit activity of Purkinje cells significantly correlated with that of single-unit activity of pyramidal cells in the cerebral cortex, we found that more pairs were observed for S1 than M1 (i.e., 11% versus 5%), (Fig. 2f, g). In addition, the magnitude of the correlations of those pairs was significantly higher for S1 than M1 ($p = 0.0496$; unpaired TTEST). In line with these results, the mutual information between spikes of Purkinje cells and S1 neurons is much higher than that between Purkinje cells and M1 neurons (Fig. 2h, $p = 0.00003$, Mann−Whitney $U$-test). Moreover, when we tested whether the phase relation between whisker movement and S1 activity was different from that between whisker movement and M1 or cerebellar activity (Fig. S5), the whisker - S1 phase relation was comparable to that of whisker - M1 ($p = 0.1827$; paired TTEST), but different from that between whisker - cerebellar activity ($p = 0.0002$; paired TTEST), highlighting the tight temporal dynamics of encodings in S1 and the cerebellum. We conclude that Purkinje cell activity of the cerebellar

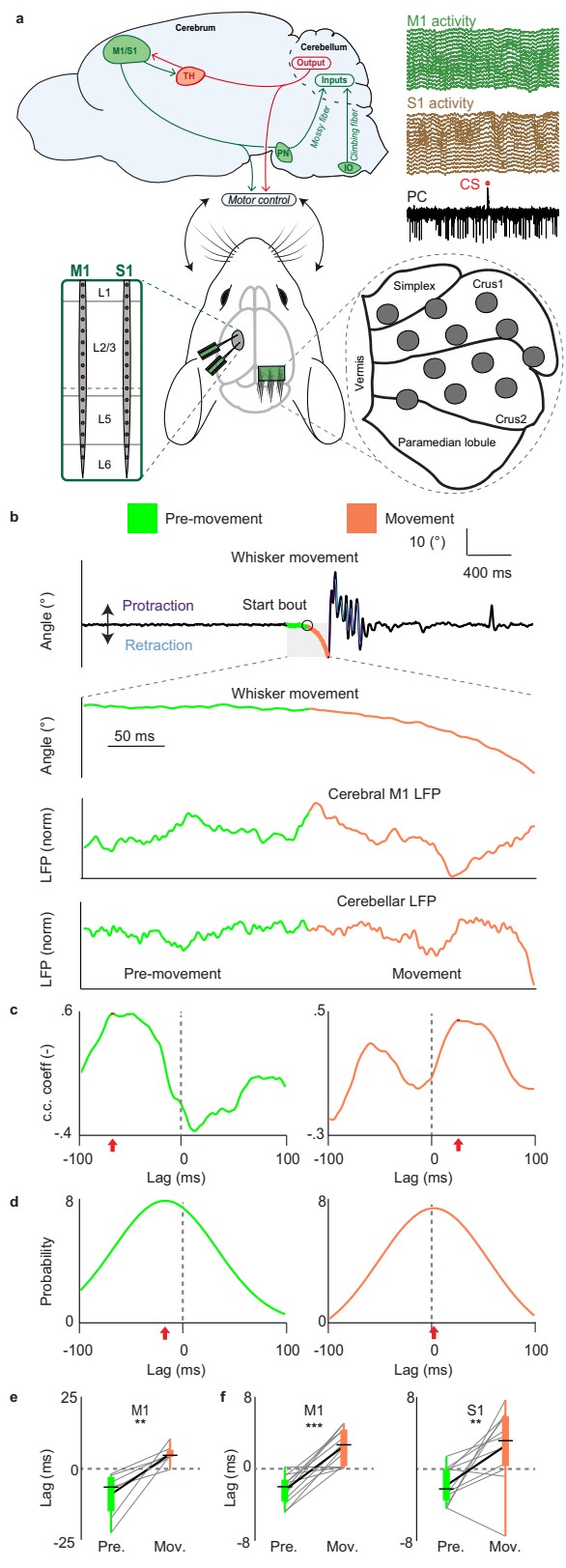

**Fig. 1 | The phase of cerebrocerebellar communication flips from planning to movement. a** Top panel-left: schematic of the main anatomical pathways connecting cerebral and cerebellar cortices. The cerebellum projects via an ascending pathway (in red) to the thalamus (TH) and from there to the primary motor (M1) and somatosensory (S1) cortex. M1 and S1 in turn project via a descending pathway (in green) back to the cerebellum using the pontine nuclei (PN) as their main intermediate hub, giving rise to mossy fibers, closing the loop. Lower panel: schematic of the silicon probes used in M1 and S1 (16 channels each) and the quartz-coated platinum/tungsten Thomas Recording electrodes placed in a 4 ×3 matrix over the cerebellar crus regions. Top panel-right: 16 exemplary raw traces of M1 and S1 as well as 1 single-unit recording from a Purkinje Cell (PC). The downward deflections of the latter trace are the so-called simple spikes, which are modulated by the mossy fiber pathway, whereas the upward deflection is a complex spike (red dot) triggered by the climbing fiber pathway derived from the inferior olive (IO). **b** Upper panel shows an example of a whisker movement with the 200 ms before and after the onset of movement highlighted in green and orange, respectively. Lower panels, enlarged whisker trace during the 200 ms before (green) and after (orange) onset of the movement (top) with an exemplary local field potential (LFP) signal from M1 (middle) and one from the cerebellar crus regions (bottom). **c** Cross-correlations of the cerebral and cerebellar LFP signals shown in panel b during the 200 ms before the onset of movement (left panels) and after the onset of movement (right panels). The dashed line at zero of the x-axis indicates that there is no phase difference between the correlated cerebral and cerebellar LFP signals. The red arrows indicate the relative time of the highest correlations between cerebral and cerebellar LFP signals (defined here as "phase relation" of correlation) before and after the individual movement onset shown in **b**. The data imply that cerebral activity is leading and lagging cerebellar activity before and after movement, respectively. **d** For this exemplary pair of LFP signals, the phase relation relative to all the movement onsets are shown as a probability density plot. The red arrows indicate the mean phase relation for this exemplary pair of LFP signals relative to only one combination of one cerebellar and one cerebral M1 recording channel. **e** For the combinations of all the cerebellar and cerebral LFP signals with a significant cross-correlation (i.e., with a Z-score > 2; see also Methods section), the mean phase relation values were averaged for cerebellar channels and used for further analysis. The thin gray lines represent the mean phase relation of the 7 significant cross-correlations between all M1 and one cerebellar channel for one exemplary mouse. The boxplot shows interquartile ranges and the thin black horizontal lines represent the medians during the Pre-movement (green) and Movement (orange) periods, respectively. The thick black line shows the mean values of all combinations of M1 and cerebellar channels, highlighting the difference between the Pre-movement and Movement stage (*p* = 0.007; Two sided paired TTEST). The whiskers indicate the entire range. **f** Left: the mean phase relation of all combinations of all M1 and cerebellar channels is shown for each of the 12 mice (*p* = 0.00005; Two sided paired TTEST). Right: similar to the left, but for all combinations of all S1 and cerebellar channels (*p* = 0.005; Two sided paired TTEST). When we used the phase relation of only significant cross-correlations (i.e., with a Z-score > 2; see also the Methods section), the results were similar as when no selection was applied. The bounds of the box indicate the interquartile range, the whiskers indicate the entire range, the black horizontal thin lines represent the medians, and the thicker black line represents the means.

between cerebral cortex and cerebellum in a bidirectional fashion via a main descending pathway including the pons and via a main ascending pathway including the thalamus[1–7]. Even though this loop is indeed one of the most prominent loops in the mammalian CNS[1–4,6,7], it is not clear to what extent this loop by itself is sufficient to explain all the dynamic, stage-dependent relationships we observed in our LFP and spike recordings. This question is particularly important as there are several side-loops that may also play a role in controlling motor behavior; these include for example the connections with the striatum or the meso-diencephalic junction, both of which are also connected with cerebral cortex and cerebellum, yet through parallel routes that are not critical to close the main continuous cerebrocerebellar loop[10–12,34–36]. To find out whether the interactions between the cerebral and cerebellar cortices through their main descending and ascending pathways are sufficient to explain their dynamics in electrophysiological activity during preparation and execution of movements, we built a neurobiologically plausible computational model of these cortices and their main intermediary

crus areas correlates in general more with S1 than M1 activity. This result is in line with the idea that cerebellar activity encodes a sensory representation of the ongoing movement.

## Modeling cerebrocerebellar dynamics

As highlighted above, our hypotheses and experiments are based on the assumption that the brain regulates motor behavior by relaying signals

**Table 1 | Summary statistics**

| Figure panel | test | *p*-value | degrees of freedom | *t*-value | effect size | Sided |
|---|---|---|---|---|---|---|
| Fig. 1e | paired t-test | 0.00700 | 6.00 | –3.00 | –1.50 | 2 |
| Fig. 1f left M1 | paired t-test | 0.00005 | 11.00 | –6.00 | –1.85 | 2 |
| Fig. 1f right S1 | paired t-test | 0.00512 | 11.00 | –6.00 | –1.01 | 2 |
| Fig. 2b left | paired t-test | 0.03228 | 11.00 | –2.05 | –0.59 | 1 |
| Fig. 2b right | paired t-test | 0.01992 | 11.00 | –2.00 | –0.79 | 2 |
| Fig. 2c | paired t-test | 0.00002 | 11.00 | –7.00 | –2.10 | 2 |
| Fig. 2f top | unpaired t-test | 0.00308 | 115.00 | –3.02 | –0.57 | 2 |
| Fig. 2f bottom | unpaired t-test | 0.04967 | 6.08 | –2.00 | –1.57 | 2 |
| Fig. 2 h | Mann-Whitney U-test | 0.00003 | 137.00 | –4.00 | –0.80 | 2 |
| Fig. 3b left | paired t-test | 0.00000 | 9.00 | –38.00 | –12.15 | 2 |
| Fig. 3b right | paired t-test | 0.00000 | 9.00 | –40.00 | –12.75 | 2 |
| Fig. 3c left | paired t-test | 0.00445 | 9.00 | –3.00 | –1.19 | 2 |
| Fig. 3c right | paired t-test | 0.00012 | 9.00 | –6.00 | –2.04 | 2 |
| Fig. 3d | paired t-test | 0.02964 | 9.00 | –2.00 | –0.68 | 1 |
| Fig. 4c top left | paired t-test | 0.00293 | 9.00 | –4.04 | –1.28 | 2 |
| Fig. 4c top right | paired t-test | 0.00000 | 9.00 | 40.00 | 12.74 | 2 |
| Fig. 4c bottom left | paired t-test | 0.00659 | 9.00 | –3.00 | –1.11 | 2 |
| Fig. 4c bottom right | paired t-test | 0.22614 | 9.00 | 1.00 | 0.41 | 2 |
| Fig. 4f top left | paired t-test | 0.00894 | 5.00 | –3.00 | –1.42 | 1 |
| Fig. 4f top right | paired t-test | 0.17183 | 5.00 | –1.05 | –0.43 | 1 |
| Fig. 4f bottom left | paired t-test | 0.04343 | 5.00 | –2.00 | –0.87 | 1 |
| Fig. 4f bottom right | paired t-test | 0.32672 | 5.00 | 0.00 | 0.19 | 1 |
| Fig. 4g | paired t-test | 0.04061 | 5.00 | –2.00 | –1.12 | 2 |
| Fig. 4h | paired t-test | 0.60483 | 5.00 | 0.00 | –0.23 | 2 |
| Fig. 5c top left | paired t-test | 0.31464 | 9.00 | 0.00 | 0.37 | 2 |
| Fig. 5c top right | paired t-test | 0.00000 | 9.00 | 58.00 | 17.47 | 2 |
| Fig. 5c bottom left | paired t-test | 0.35397 | 9.00 | 0.00 | 0.34 | 2 |
| Fig. 5c bottom right | paired t-test | 0.00000 | 9.00 | 53.00 | 16.06 | 2 |
| Fig. 5f top left | paired t-test | 0.40817 | 8.00 | 0.00 | 0.08 | 1 |
| Fig. 5f top right | paired t-test | 0.03243 | 8.00 | 2.00 | 0.71 | 1 |
| Fig. 5f bottom left | paired t-test | 0.17068 | 8.00 | –1.01 | –0.34 | 1 |
| Fig. 5f bottom right | paired t-test | 0.29552 | 8.00 | 0.00 | –0.19 | 1 |
| Fig. 5g | paired t-test | 0.15737 | 8.00 | –1.00 | –0.52 | 2 |
| Fig. 5h | paired t-test | 0.03835 | 8.00 | 2.00 | 0.83 | 2 |
| Fig. 6c | paired t-test | 0.00000 | 9.00 | 23.00 | 7.40 | 2 |
| Fig. 6f | paired t-test | 0.01135 | 5.00 | 3.00 | 1.33 | 2 |
| Fig.6i | paired t-test | 0.04730 | 5.00 | 2.00 | 1.07 | 2 |
| Fig. 6k ChR | paired t-test | 0.38575 | 4.00 | 0.00 | –0.14 | 2 |
| Fig. 6k HR | paired t-test | 0.18108 | 4.00 | 1.03 | 0.46 | 2 |
| Figure S2a left | paired t-test | 0.00000 | 15.00 | –8.00 | –2.10 | 2 |
| Figure S2a right | paired t-test | 0.00000 | 15.00 | –15.00 | –2.87 | 2 |
| Figure S2c left | paired t-test | 0.00045 | 7.00 | –3.00 | –1.30 | 2 |
| Figure S2c right | paired t-test | 0.77252 | 7.00 | 0.00 | –0.10 | 2 |
| Figure S4a left | Mann–Whitney U-test | 0.00000 | 62.00 | 4.00 | 0.63 | 2 |
| Figure S4a right | Mann–Whitney U-test | 0.00144 | 75.00 | 2.00 | 0.34 | 2 |
| Figure S4b left | Mann–Whitney U-test | 0.00000 | 62.00 | 5.00 | 0.70 | 2 |
| Figure S4b right | Mann–Whitney U-test | 0.00000 | 75.00 | 9.00 | 1.11 | 2 |
| Figure S4c left | paired t-test | 0.00509 | 56.00 | 2.00 | 0.35 | 2 |
| Figure S4c right | paired t-test | 0.16857 | 75.00 | –1.00 | –0.16 | 2 |
| Figure S4d | paired t-test | 0.02326 | 55.00 | 2.04 | 0.27 | 2 |
| Figure S4e | paired t-test | 0.10898 | 67 | 1 | –0.15 | 2 |
| Figure S5c left | paired t-test | 0.13295 | 11.00 | 1.00 | 0.47 | 2 |
| Figure S5c middle | paired t-test | 0.18269 | 11.00 | –1.00 | –0.41 | 2 |
| Figure S5c right | paired t-test | 0.00022 | 11.00 | 5.00 | 1.56 | 2 |
| Figure S6a top left | paired t-test | 0.00222 | 11.00 | –3.00 | –1.14 | 2 |

**Table 1 (continued) | Summary statistics**

| Figure panel | test | *p*-value | degrees of freedom | *t*-value | effect size | Sided |
|---|---|---|---|---|---|---|
| Figure S6a top right | paired t-test | 0.02982 | 11.00 | −2.00 | −0.72 | 2 |
| Figure S6a bottom left | paired t-test | 0.62406 | 11.00 | 0.00 | 0.15 | 2 |
| Figure S6a bottom right | paired t-test | 0.03125 | 11.00 | −2.00 | −0.71 | 2 |
| Figure S6c top left | paired t-test | 0.01880 | 11.00 | −2.00 | −0.79 | 2 |
| Figure S6c top right | paired t-test | 0.01275 | 11.00 | −2.00 | −0.86 | 2 |
| Figure S6c bottom left | paired t-test | 0.01808 | 11.00 | 2.00 | 0.80 | 2 |
| Figure S6c bottom right | paired t-test | 0.00070 | 11.00 | 4.00 | 1.34 | 2 |

The table reports relevant values of all tests performed in this manuscript. The rightmost column indicates whether the test is one or two–sided test.

hubs of the descending and ascending pathways, while discarding the side-loops through the striatum and mesodiencephalic junction (Fig. 3a). More specifically, in our model, M1 and S1 project via the descending path to the pontine nuclei, from which the mossy fibers arise that innervate populations of granule cells providing excitatory input to the Purkinje cell population[37]. Purkinje cells inhibit the cerebellar nuclei, which in turn modulate activity in the ventrolateral thalamus (VL) and medial posterior thalamic nucleus (Pom), either directly via an ascending, excitatory projection or indirectly via an ascending, inhibitory projection from the zona incerta (ZI)[38]. The VL projects to M1, while the Pom innervates both M1 and S1, together closing the loop between cerebrum and cerebellum. In our model, M1 and S1 are modeled as simplified laminar circuits with superficial and deep layers that interact bidirectionally, as previous work revealed the importance of laminar-dependent interactions to explain M1-S1 dynamics[9,39,40]. We modeled the dynamics of the firing rates of these cortical regions and hubs using population-based averages or neural-mass models constrained by anatomically-informed connectivity (for details, see Methods section). Both cerebral and cerebellar cortex receive noisy sensory input, with S1 having a relatively high level of common noisy input with the cerebellar network, reflecting spontaneous activity of sensory afferents[25]. Moreover, during the Pre-movement and Movement conditions, we increased and decreased the intensity of preparatory ramping signals to M1, respectively. These temporary modulations align well with recordings done in prefrontal cortex[41] and cerebellar nuclei[3,42], as well as with the concept of the higher cortical areas providing stage-dependent signals to the cerebellum for generating sensorimotor predictions[17].

In agreement with our experimental findings, the simulated cross-correlations between M1 and Purkinje cells displayed an early peak in the phase relation during the Pre-movement stage, indicating that the cortex was leading the cerebellar activity during this preparatory stage (Fig. 3b, upper left panel). According to the model, this M1-led cerebellar activity was due to the strong Pre-movement signal in M1 propagating to the Purkinje cells via the descending pathway (Fig. 3a). Conversely, our simulations indicated that Purkinje cells were leading M1 during the Movement stage (Fig. 3b, upper right panel). Given that the only difference with the Pre-movement phase in the model is the presence of the ramping signal in M1, our results suggest that the cortico-cerebellar lead can be inverted by simply regulating preparatory signals in motor cortex, without further alterations in the cerebellar or thalamic networks. The difference between the cerebrocerebellar phase relations of the Pre-movement stage compared to those of the Movement stage was significant for both M1 and S1 (in both cases, *p* < 0.001; paired TTEST). To find out whether there was a difference between M1 and S1 in this respect, we next quantified the correlation between the activity of Purkinje cells and that of M1 and S1 (Fig. 3c). As predicted, calculation of Pearson's correlation R revealed a positive correlation that was stronger for S1 than M1, primarily due to the joint sensory input received by S1 and the cerebellum. Moreover, in line with our experimental LFP data, this held true for both the Pre-movement (*p* = 0.004; paired TTEST) and the Movement (*p* = 0.0001; paired TTEST) period, and the correlation between activity of the

Purkinje cells and that of S1 was stronger during the Movement than Pre-movement period (*p* = 0.0296; paired TTEST), (Fig. 3d). Given that the preparatory signals in M1 during the Pre-movement phase would indirectly introduce a higher level of randomness with both S1 and the cerebellum, leading to a decrease in their correlation, these modeling data provide mechanistic insight as to how the dynamics may come about. We conclude that our closed-loop model of cerebrocerebellar communication via direct descending and ascending paths, but without side-loops that engage the striatum and/or mesodiencephalic junction, is sufficient to accurately replicate our experimental recordings on the phase relations between activity in M1, S1 and the cerebellar crus regions during both the preparation and execution of naturally occurring whisker movements.

## Role of the descending pathway

After assessing how well the model reproduced the basic features of the cerebrocerebellar phase dynamics, we aimed to generate and test new predictions on the role of the descending pathway during both the Pre-movement and Movement stage. We interrogated the model on the impact of inhibiting the pons, which is the main intermediary hub of the descending pathway (Fig. 4a). Our model showed that the phase lead of activity of the cerebral cortex with respect to that in the cerebellum during the Pre-movement stage should turn into a significant lag following this manipulation (Fig. 4b, top panel). Indeed, according to our model, inhibition of the pons leads to such a phase shift for both M1 (M1-cerebellum *p* = 0.003, paired TTEST) and S1 (S1-cerebellum *p* = 0.007) during the Pre-movement stage (Fig. 4c, left panel), whereas there is a small (but significant) phase shift in the opposite direction or no significant shift at all during the Movement stage for M1 and S1, respectively (for M1-cerebellum *p* < 0.001 and for S1-cerebellum *p* = 0.226; paired TTEST), (Fig. 4c, right panel). As a consequence, our model simulations predict that inhibition of the descending pathway via the pons should differentially affect the Pre-movement and Movement stage.

To test the predicted role of the descending pathway we set out to do in vivo LFP recordings following a 200 ms pulse of optogenetic manipulation of the pontine neurons that receive input from either M1 or S1 (Fig. 4d, Fig. S6a). We selectively targeted the descending cerebro-ponto-cerebellar pathway by injecting transneuronal Cre virus in M1 or S1 of the cerebral cortex and Cre-dependent stGtaCR2 in the pons (Fig. 4e), and we subsequently compared brain activity around movement onset with and without light stimulation. In line with the model predictions, we found that pontine inhibition significantly affected the phase relations between cerebral and cerebellar activity during the Pre-movement stage (for M1-cerebellum *p* = 0.009 and for S1-cerebellum *p* = 0.043; paired TTEST), whereas it did not change those of the Movement stage (for M1-cerebellum *p* = 0.258 and for S1-cerebellum *p* = 0.327; paired TTEST), (Fig. 4f–h). Interestingly, inhibition of the descending pathway significantly increased the probability of the initiation of a new whisking bout (*p* = 0.041; paired TTEST), (Fig. 4g). This may well be explained by disinhibition of the main whisker pre-motor area, i.e., the vibrissae-related intermediate reticular formation, that normally inhibits vibrissa facial motoneurons,

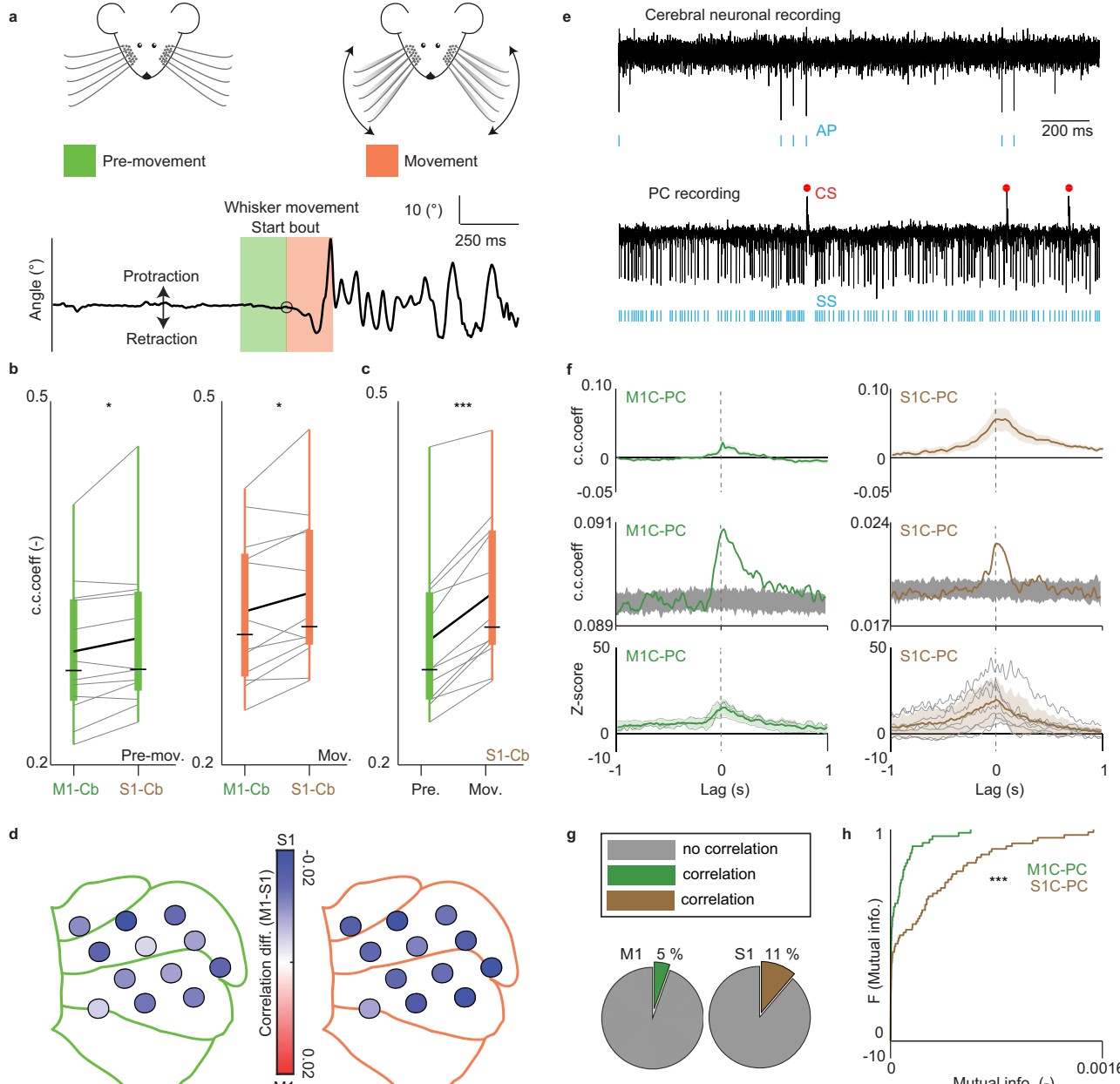

**Fig. 2 | Purkinje cell activity correlates more with sensory than motor cortical activity. a** Schematic of the two different time windows (200 ms) we considered: the Pre-movement (green) and Movement (orange) stage. **b** Comparison of the correlation strength between cerebellum (CB) and M1 versus CB and S1 during Pre-movement ($p = 0.032$, One sided paired TTEST), and Movement ($p = 0.0199$, Two sided paired TTEST). Each line represents the average of peaks of the cross correlograms for each mouse ($n = 12$). The bounds of the box indicate the inter-quartile range, the whiskers indicate the entire range, the black horizontal thin lines represent the medians, and the thicker black line represents the means. **c** Comparison of the correlation strength between CB and S1 during Pre-movement versus Movement ($p < 0.001$, paired Two sided TTEST, $n = 12$ mice). The bounds of the box indicate the interquartile range, the whiskers indicate the entire range, the black horizontal thin lines represent the medians, and the thicker black line represents the means. **d** Average difference in correlation strength of CB and M1 minus CB and S1, per cerebellar location. A negative difference (blue color) indicates that the cerebellar correlation with S1 is stronger than with M1. **e** Example of a single-unit S1 neuron (top, with the action potentials APs indicated by blue dashes) and a simultaneously recorded Purkinje cell (PC), (bottom, with the complex spikes (CSs) and simple spikes (SSs) indicated by red dots and blue dashes, respectively). **f** Top left: Mean of cross-correlation of all 55

pairs of SSs from PCs and the single-unit activity of M1 neurons in green. Shaded areas represent SEM. Top right: mean cross-correlation of all 62 pairs of SSs from PCs and the single-unit activity of S1 neurons in brown. Note that overall S1-PC cross-correlation is stronger than C1-PC cross-correlation ($p = 0.003$; Two sided unpaired TTEST). Middle left: cross-correlation of the SSs of a PC and the single-unit activity of an M1 neuron in green superimposed on 100 correlations obtained through shuffling the M1 activity randomly (gray). Middle right: similar to the left panel, but for SSs of a PC and a neuronal recording from S1 (same cell as in E). Lower panels: all cross-correlations that significantly exceeded the range of the randomized data (3 out of the 55 pairs for M1 and 7 out of the 62 pairs for S1, thin gray lines) with their mean (colored thicker line) and standard deviation (shaded area) of the Z-scores. **g** Percentage of pairs of PC-cortical single units for which the cross-correlation crossed the range of the variability that could be expected by chance (see also Methods). The percentage of significantly correlating PC-M1 pairs (in green) is about half of the PC-S1 pairs (in brown). **h** Comparison of the cumulative distribution functions of the mutual information between pairs of Purkinje cell-M1 neuron (green) and pairs of Purkinje cell-S1 neuron (brown). Note that the mutual information between spikes of Purkinje cells and S1 neurons is much higher than that between Purkinje cells and M1 neurons ($p = 0.00003$, Two sided Mann–Whitney U-test).

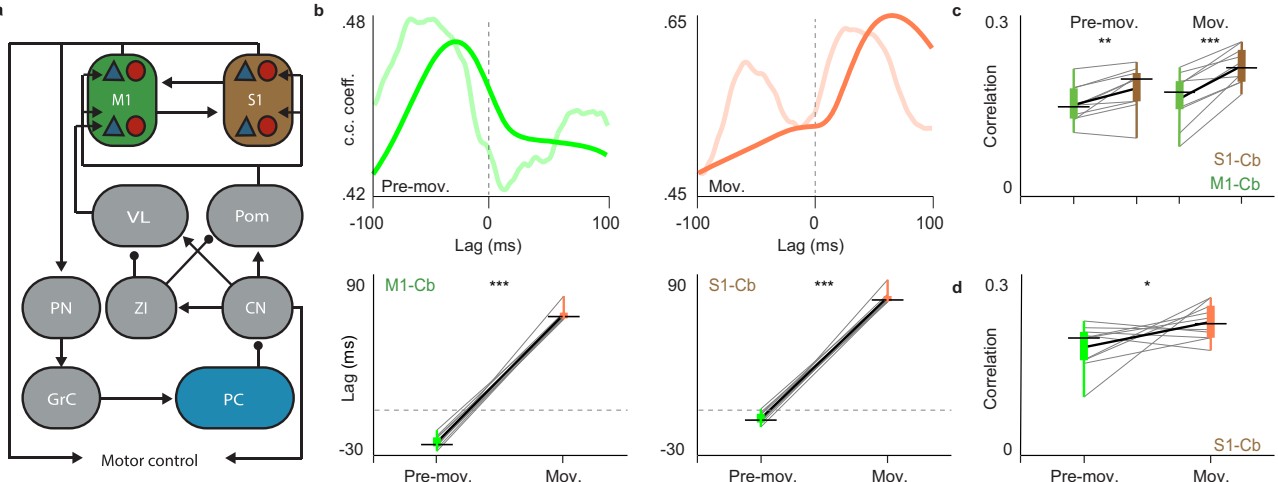

**Fig. 3 | Modeling cerebrocerebellar dynamics. a** Schematic of the modeled network, which includes primary motor cortex (M1), primary somatosensory cortex (S1), pontine nucleus (PN), granule cells (GrC), Purkinje cells (PC), cerebellar nuclei (CN), zona incerta (ZI), ventrolateral thalamus (VL) and medial posterior nucleus of the thalamus (Pom). **b** Cross-correlation between Purkinje cell activity and M1 activity in the Pre-movement epoch (green) and Movement epoch (orange). The solid lines represent the cross-correlograms of the modeled data, while the shaded lines are the cross-correlograms of the experimental data shown in Fig. 1c. The phase relation of the peak of the maximum correlation is shown in the bottom panels for M1 (left, $p = 2.71305E\text{-}11$; Two sided paired TTEST) and S1 (right, $p = 1.76631E\text{-}11$; Two sided paired TTEST) for 10 independent model simulations.

These data suggest that our modeled closed-loop network is sufficient to explain the lag and lead of the cerebellar activity with respect to that in cerebral cortex during the Pre-movement and Movement stage, respectively. **c** The level of the M1 - PC correlation (Pearson's R) is compared with that of the S1 - PC correlation for 10 independent simulations during the Pre-movement ($p = 0.004$; Two sided paired TTEST) and Movement condition ($p < 0.001$; Two sided paired TTEST). Cb refers to cerebellum. **d** Comparison of the correlation strength between cerebellum and S1 during Pre-movement versus Movement for the for 10 independent model simulations ($p = 0.0296$; One sided paired TTEST). The bounds of the box indicate the interquartile range, the whiskers indicate the entire range, the black horizontal thin lines represent the medians, and the thicker black line represents the means.

controlling rhythmic whisking[28,43]. Equally interesting from a neuroanatomical perspective is the finding that inhibition of the pontine nuclei with optogenetics particularly affected Purkinje cells located in the lateral part of crus 2 (Fig. S6c), which is the region in which the correlation between simple spike activity and whisker position is highest[24]. Together, these results indicate that the phase relation between activity in the cerebral cortex and cerebellum during the Pre-movement stage reflect the activity of the descending cerebro-ponto-cerebellar pathway. We propose that the signals carried by this pathway provide a copy of the motor commands generated in the cerebral cortex to the cerebellum, which may help to facilitate movement preparation[44].

## Role of the ascending pathway

After probing the descending pathway, we next generated and tested new predictions on the role of the ascending pathway during both the Pre-movement and Movement stage. We interrogated the model on the impact of inhibiting the VL, which is the main intermediary thalamic hub of the ascending pathway from cerebellum to cerebral cortex (Fig. 5a). Our model showed that VL inhibition is sufficient to affect the cerebrocerebellar dynamics during the Movement stage (for cerebellum-M1 $p < 0.001$ and for cerebellum-S1 $p < 0.001$; paired TTEST), but not during the Pre-movement stage (for cerebellum-M1 $p = 0.315$ and for cerebellum-S1 $p = 0.354$; paired TTEST), (Fig. 5b, c). Interestingly, even though VL does not project directly to S1, VL inhibition did not only significantly perturb the temporal dynamics between cerebellum and M1 during the Movement stage, but also that between cerebellum and S1, presumably reflecting the prominent intercortical connections between M1 and S1 in our model. Overall, this solidifies our intuition on the importance of VL thalamus in cerebellar control of S1 and especially M1.

To test these modeled predictions experimentally in vivo, we targeted the ascending cerebellar-thalamic pathway by injecting transneuronal Cre virus in the lateral cerebellar nucleus and Cre-dependent stGtaCR2 in the VL of the thalamus (Fig. 5d, e, see also

Fig. S6). In line with our model, we found that a 200 ms pulse of optogenetic inhibition of this hub in the ascending pathway affected the phase lead of activity in the cerebellum with respect to that in M1 during the Movement stage (cerebellum-M1 $p = 0.032$; paired TTEST), but not during the Pre-movement stage (cerebellum-M1 $p = 0.408$; paired TTEST), (Fig. 5f–h). Opposite to inhibition of the PN of the descending pathway (Fig. 4g, h), we did not observe an increase in probability of initiation of new whisker bouts ($p = 0.1574$; paired TTEST), but we did find a significant change in the power-spectrum of the movement components ($p = 0.0383$; paired TTEST), (Fig. 5g, h). In addition, as predicted, optogenetic inhibition of the thalamus did not significantly alter the phase between activity in the cerebellum and that in S1 during the Pre-movement stage (cerebellum-S1 $p = 0.171$; paired TTEST). However, optogenetic inhibition of the thalamus also did not significantly alter the phase between activity in the cerebellum and that in S1 during the Movement stage (cerebellum-S1, $p = 0.295$; paired TTEST), which diverged from our modeled prediction.

These latter data raise the possibility that changes in cerebellar activity have in fact more impact on M1 than S1 when the ascending pathway is engaged in the Movement stage, despite the finding that the general correlations of cerebrocerebellar activity during spontaneous behavior are strongest for S1 (Fig. 2c). We therefore interrogated again the ascending part of the model by comparing the cortical M1 and S1 responses to modulation of Purkinje cells (Fig. 6). Our model indicated that the impact of Purkinje cell activation would indeed be bigger on M1 than S1 ($p < 0.001$; paired TTEST), (Fig. 6b, c). Thus, even though correlations between the cerebellum and S1 are stronger due to the shared sensory inputs (as shown in Fig. 3c), according to our modeling assumptions the ascending pathway targets M1 more strongly than S1. To examine this potential difference experimentally in vivo, we specifically stimulated Purkinje cells, exploiting a mouse model (L7-Ai27, ChR2) that expresses channelrhodopsin 2 under the L7-vector (Fig. 6d). We focused our stimulation on the lateral portion of crus 1, which was the area where inhibition of VL had its major impact (Fig. S6d). When we stimulated the Purkinje cells, the

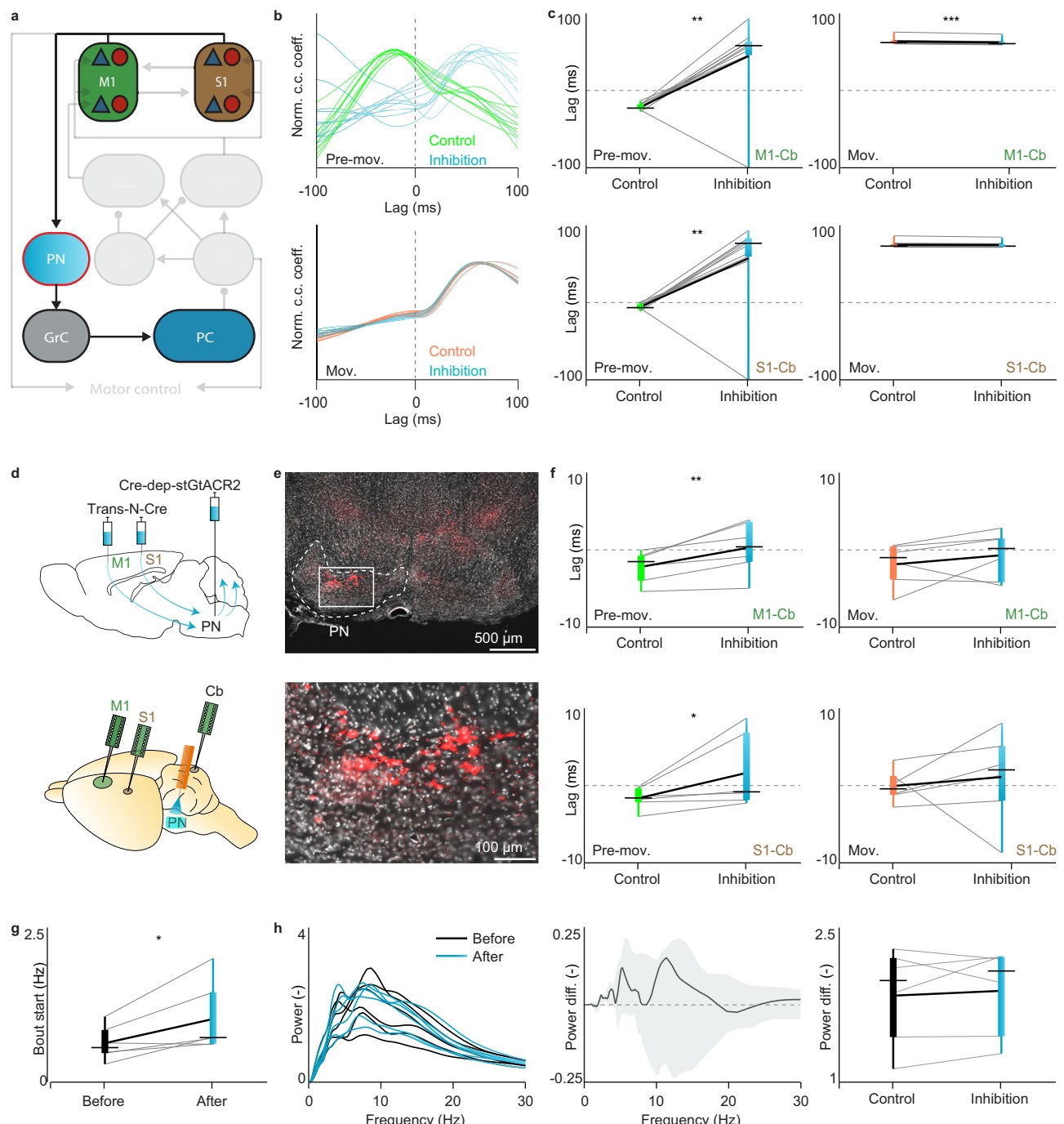

magnitude of the LFP peak responses in the first 100 ms after stimulus onset was bigger in M1 than in S1 ($p = 0.011$; paired TTEST; Fig. 6e, f), despite the fact that the onset of the responses was similar in both areas ($p = 0.386$; paired TTEST; Fig. 6k). Moreover, these effects were most prominently visible in the recordings of the deeper located channel outlets (Fig. 6j), which is in line with literature[29–31] and our data presented above (Fig. S3). We next investigated the specific impact of inhibition of Purkinje cells in L7-Ai39 mice that express halorhodopsin (HaloR), (Fig. 6g–k). For the majority of the recorded channels the polarity of the effect following this inhibition was opposite to that following excitation in both M1 and S1 (in both cases $p < 0.000001$; paired TTEST), with again the deeper located channels showing the most prominent impact (Fig. 6j). In addition, here too, the moments of onset of the inhibitory responses in M1 and S1 were comparable ($p = 0.176$; paired TTEST; Fig. 6k), while the magnitude of the

responses in M1 was greater than that in S1 ($p = 0.047$; paired TTEST), (Fig. 6g–i). The observation that bidirectional manipulation of the Purkinje cells induced opposing effects in cerebral cortex underscores the specificity of our findings. Together, our results highlight that the impact of Purkinje cells on cerebral cortical activity is particularly prominent in M1 and that this impact can be associated with the Movement stage, when cerebellar output may adjust ongoing movement.

## Discussion

Our experimental and modeling data indicate that neocortical signals predominantly lead cerebellar neuronal activity during movement preparation and that this direction of flow largely reverses after movement initiation. These temporal dynamics were similar for the primary motor and primary sensory cortex, and optogenetic

**Fig. 4 | Stage-dependency of the cerebrocerebellar communication upon manipulation of the descending pathways. a** Schematic of the interference of the descending pathway via inhibition of the pontine nuclei (PN) in the modeled network. **b** Cross-correlograms of 10 independent control simulations (green) and 10 simulations in which inhibition was applied in the modeled network to the PN (blue) during the Pre-movement (top) or Movement (below) stage. Control traces are here indicated in orange. Normalized cross-correlograms are indicated by norm. c.c. coeff. **c** The phase relations for the maximum cross-correlations as obtained with modeled manipulations are shown for the control versus the PN inhibition condition. Left, during the Pre-movement epoch, the negative values of the phase relation in the control condition become positive when PN inhibition is applied (top, PC-M1 $p = 0.003$; below, PC-S1 $p = 0.007$ Two sided paired TTEST). Right, during the Movement epoch, the positive values of the phase relation in the control condition remain positive when PN inhibition is applied, but they become slightly lower for the M1-PC correlation (top, PC-M1 $p = 1.78468E-11$; below, PC-S1 $p = 0.226$; Two sided paired TTEST). The bounds of the box indicate the interquartile range, the whiskers indicate the entire range, the black horizontal thin lines represent the medians, and the thicker black line represents the means. **d** Schematic of the viral approach to target PN neurons receiving projections from M1 and S1 (left). Schematic of the recordings in M1, S1 and cerebellar cortex, while manipulating PN activity. **e** Neurons labeled with red fluorescent protein within the pontine nuclei.

The ventral part of a coronal section at about −4.24 mm from bregma is shown above and the magnification of the white rectangle is magnified below. Similar neurons labeling was detected in 6 mice. **f** Similar to **c**, but for the experimental data following optogenetic manipulation ($n = 6$ mice). Here, the "control" refers to the same mice with stGtACR2 expression, but with the LED light off. Inhibiting the PN, the phase relation for the maximum cross-correlation changed during the pre-movement epoch (left, PC-M1 $p = 0.009$, PC-S1 $p = 0.043$; One sided paired TTEST), but not during the movement epoch (right, PC-M1 $p = 0.258$, PC-S1 $p = 0.327$; One sided paired TTEST). The bounds of the box indicate the interquartile range, the whiskers indicate the entire range, the black horizontal thin lines represent the medians, and the thicker black line represents the means. **g** Rate of self-initiated whisking bouts in the epoch before and after the stimulus initiation ($p = 0.041$; Two sided paired TTEST). **h** The power spectra of the whisker movement in the 500 ms epoch before (black) and after (cyan) the stimulus onset are shown for all mice (left, $n = 6$). The median of the power spectrum differences (before minus after the stimulus) is shown with the interquartile range (middle). Comparison of the mean power between 3 and 10 Hz (right, $p = 0.605$; Two sided paired TTEST). The bounds of the box indicate the interquartile range, the whiskers indicate the entire range, the black horizontal thin lines represent the medians, and the thicker black line represents the means.

manipulation of the descending and ascending paths between these cerebral areas and the cerebellar hemisphere confirmed the stage-dependent dynamics predicted by the model.

Feedforward theories of the cerebrocerebellum suggest that the cerebellum receives a copy of the motor command generated in the cerebral cortex so as to predict the next sensory state[14,17]. Our findings are compatible with the concept of such a forward model in that the increases of activity in the cerebral cortex slightly, but consistently and significantly, precede those of the cerebellar cortex during movement preparation. As the feedforward model requires feedback during the actual execution of the movement to optimize its future predictions[45], our data obtained during motor execution also align well in that the main flow of signals during this stage turns out to be directed from the cerebellum to the primary motor cortex. Since we show that the Purkinje cell activity during well-executed, spontaneous movements correlates better with the activity of putative pyramidal cells in the sensory cortex than with that in the motor cortex, our results are also consistent with the possibility that the feedback signals during motor execution are being used to improve the prediction of the sensorial state[17]. In this respect, it is interesting to note that the receptive fields of Purkinje cells in the crus regions, where we have obtained our data, can also be linked to specific whisker pads, akin the barrel-like structure in the sensory cortex[19,25,26,46,47].

If the cerebellum computes, as predicted by theoretical models[14], indeed an improved prediction of the sensorial state for subsequent events, it is attractive to speculate that the cerebellum provides an error signal when there is a mismatch between the actual and the predicted movement. Given that such signals have indeed been shown not only in the sensorimotor domains of whisking and eye movements[24,48,49], but also following cognitive visuomotor learning[24,48-51], we propose that the output of the cerebellum affects ongoing movement for online motor correction via premotor nuclei in the brainstem, that the same output is relayed via the ascending thalamic pathway to the cerebral cortex to improve subsequent motor commands[52,53], and that ultimately a related output is also fed back into the inferior olive to recalibrate the internal model processed within the olivocerebellar system itself[54,55]. For areas like visuomotor identification, touching limb movements as well as exploratory whisking, such optimization of motor control will probably also lead to improved acquisition and processing of sensory inputs[56,57], further strengthening the role of the cerebrocerebellar loop in sensorimotor coordination[9,58-60].

Excitatory and inhibitory optogenetic manipulation of the Purkinje cells resulted in opposite changes in SS firing and thereby induced LFP responses in a bidirectional fashion in most of the

recorded channels of both M1 and S1, highlighting the level of specificity with which cerebellar output can directly control sensorimotor cortices. Albeit these effects were prominent in the deeper layers, they were less clear in the superficial layers of M1 and S1. As the cerebellar-cortical projections of the whisker system target different cortical layers via different thalamic subnuclei[25,31], the intensity and specificity of cerebellar inputs may well vary across cortical layers. The fact that the amplitudes of the responses we found in the superficial layers of the sensorimotor cortex were smaller in both M1 and S1 suggests that the impact of the cerebellar output targeting layers IV, V and VI is in general more specific than that targeting layers II and III[9,25,61]. In addition, the similarity in the cerebral responses following optogenetic Purkinje cell stimulation might be facilitated by the prominent reciprocal input between M1 and S1 at the level of layer II/III, which may exert a coupling effect[61].

Our computational model provided mechanistic insights into how the cortical lead on the cerebellum during the Pre-movement phase could be inverted at the onset of the Movement phase; in principle, this swap could be evoked simply by reducing the preparatory activity in M1, without further critical alterations in the cerebellar or thalamic network. The model also helps to explain how the cerebellum may display stronger correlations with S1, while having a stronger impact on M1, particularly during the Movement stage. Moreover, the predictions of our model regarding the significant effects upon inactivation of the ascending and descending pathways for Movement and Pre-movement phases, respectively, were largely confirmed by experimental testing, further validating our theoretical framework. Yet, as a neural-mass network, our modeling approach is limited in its capacity to translate neural activity into concrete computations encoding specific movements. Recent modeling work based on cortico-cerebellar recurrent neural networks provides an efficient approach to capture such computations[52,53], and combining these types of networks with our large-scale modeling domain may well form a promising way forward[59].

The data obtained with our computational model indicate that the closed-loop configuration of cerebrocerebellar interactions via the direct descending and ascending paths, including the central pontine and thalamic hubs, but not the parallel hubs of the striatum and/or mesodiencephalic junction, is sufficient to accurately explain the electrophysiological dynamics between activity in M1, S1 and the cerebellar crus regions during both the preparation and execution of spontaneous whisker movements. This raises the possibility that areas like the striatum and mesodiencephalic junction are more prominently engaged during more demanding tasks like execution of voluntary

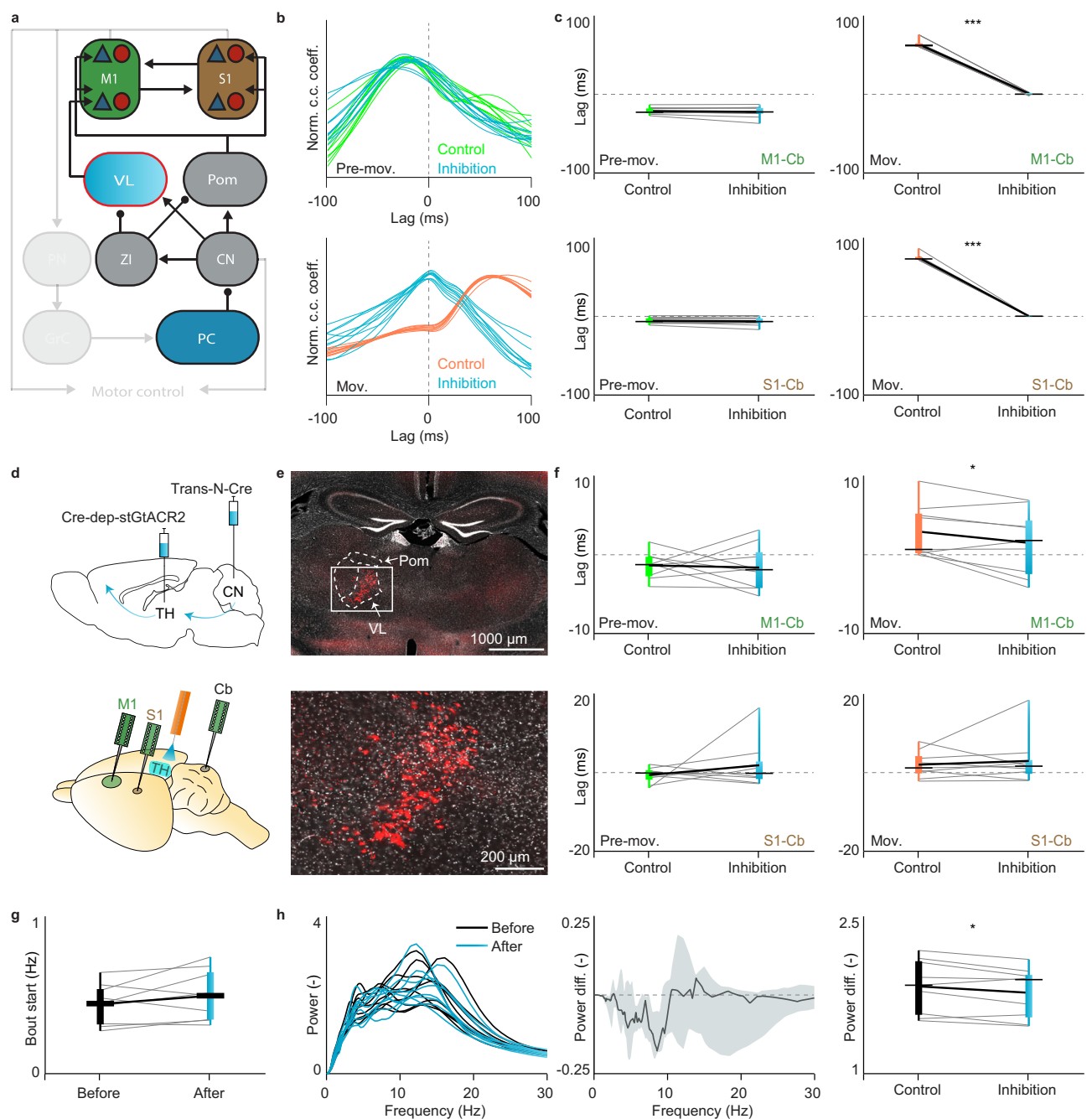

goal-directed movements, which often include sequence and delay learning[34–36]. Indeed, the basic cerebrocerebellar loop may be sufficient to coordinate innate and ethological motor behaviors, such as spontaneous whisking or licking that occur from early on[8,9], while superimposed systems may be required for more complex tasks that are acquired later in life[53]. Moreover, if the behavioral demand is more challenging, the dynamics within the cerebrocerebellar system may also vary in that the movement planning occurring in cerebral cortex may depend a priori on guiding and fine-tuning signals from cerebellum[3,45,52,53,60].

## Methods

### Mice

All mice experiments were done according to the animal guidelines of the institutional animal welfare committee of Erasmus MC and approved a priory by an independent animal ethical committee (DEC-Consult, Soest, the Netherlands) as required by Dutch law on animal experimentation. Wild-type C57BL/6 J (No. 000664), transgenic L7-Ai27D (No. 012567) and L7-Ai39 (No. 014539) mice were obtained from the Jackson Laboratory. Adult 6-34 weeks old mice were used in this study and the mice were housed individually in a 12-h light-dark cycle with food and water *ad libitum*. The ambient housing temperature was maintained at ~25.5 °C with 40-60% humidity. We used 12 mice in the correlation experiments in the neocortex and cerebellum (9 males and 3 females; age = 193 ± 62 days, mean ± standard deviation) and 35 mice for stimulation experiments in the neocortex, cerebellum, thalamus or pons (5 males and 30 females; age = 178 ± 41 days, mean ± standard deviation). After the experiment mice were euthanized by cervical dislocation.

### Surgeries

For all mice, a magnetic pedestal was placed on the skull above the bregma using Super-Bond C&B (Sun medical, Furutaka-cho, Japan) and craniotomies were made over the left hemisphere of the cerebral

**Fig. 5 | Stage-dependency of the cerebrocerebellar communication upon manipulation of the ascending pathways. a** Schematic of the interference of the ascending pathway via inhibition of the ventrolateral thalamic nucleus (VL) in the modeled network (left). **b** Cross-correlograms between M1 and PC activity in 10 independent control simulations (orange) and 10 simulations in which inhibition was applied to the VL (cyan) during the Pre-movement (top) and Movement stage (bottom). Normalized cross-correlograms are indicated by norm. c.c. coeff. **c** Phase relations of the simulations in which VL inhibition is applied ($n = 10$ simulations). In this case, the inhibition of the ascending pathway changes the phase relation of the cerebrocerebellar cross-correlation specifically during the Movement epoch (Pre-movement: PC-M1 $p = 0.315$, PC-S1 $p = 0.354$; Movement: PC-M1 $p = 6.27707E-13$, PC-S1 $p = 1.33738E-12$; Two sided paired TTEST). The bounds of the box indicate the interquartile range, the whiskers indicate the entire range, the black horizontal thin lines represent the medians, and the thicker black line represents the means. **d** Schematic of the viral approach to target the thalamic neurons receiving projections from the cerebellar nuclear neurons (top). Schematic of the recordings in M1, S1 and cerebellar cortex, while manipulating thalamic activity (bottom). **e** Neurons labeled with red fluorescent protein in the VL. The central part of a coronal section at about −1.46 mm from bregma is shown in the upper panel and the magnification of the white rectangle is in the bottom panel. Pom = medial

posterior nucleus of the thalamus. Similar neurons labeling was detected in 9 mice. **f** Optogenetic inhibition of the thalamus changed the phase relation for the maximum cross-correlation of M1 during the Movement epoch (right, PC-M1 $p = 0.032$; One sided paired TTEST), but not during the Pre-movement epoch (left, PC-M1 $p = 0.408$; One sided paired TTEST, $n = 9$ mice). This manipulation did not significantly alter the S1-cerebellar correlations (Pre-movement epoch: PC-S1 $p = 0.171$; paired One sided TTEST, Movement epoch: PC-S1 $p = 0.295$; One sided paired TTEST; $n = 9$ mice) (below). Each line represents one mouse. Here, the "control" refers to the same mice with stGtACR2 expression, but with the LED light off. The bounds of the box indicate the interquartile range, the whiskers indicate the entire range, the black horizontal thin lines represent the medians, and the thicker black line represents the means. **g** Rate of self-initiated whisking bouts in the epoch before and after the stimulus initiation ($p = 0.157$; Two sided paired TTEST). **h** The power spectra of the whisker movement in the 500 ms epoch before (black) and after (cyan) the stimulus onset are shown for all mice (left, $n = 9$). The median of the power spectrum differences (before minus after the stimulus) is shown with the interquartile range (middle). Comparison of the mean power between 3 and 10 Hz (right, $p = 0.038$; Two sided paired TTEST). The bounds of the box indicate the interquartile range, the whiskers indicate the entire range, the black horizontal thin lines represent the medians, and the thicker black line represents the means.

cortex and the right hemisphere of the cerebellum. Isoflurane Fstatana-esthesia (Pharmachemie, Haarlem, The Netherlands; 2-4% V/V in $O_2$) was maintained during the whole surgery procedure. Mice were given 5 mg/kg carprofen ("Rimadyl", Pfizer, New York, USA), 50 μg/kg buprenorphine ("Temgesic", Reckitt Benckiser Pharmaceuticals, Slough, United Kingdom), 1 μg lidocaine (AstraZeneca, Zoetermeer, The Netherlands) and 1 μg bupivacaine (Actavis, Parsippany-Troy Hills, NJ, USA) to reduce post-surgical pain. After 48 hours of recovery, mice were habituated to the recording apparatus for about 45 minutes during at least 2 daily sessions. In the recording apparatus, mice were head-fixed with the pedestal and restrained.

## Whisker tracking
The whisker movements were tracked using the BIOTACT Whisker Tracking Tool[62] in combination with custom-written code (https://github.com/elifesciences-publications/BWTT_PP) or our software tracking tool[63], (https://gitlab.com/c7859/neurocomputing-lab/whisker/whiskeras-2.0). The whisker movements were described as the average angle of all trackable whiskers per frame.

## Electrophysiology and spike sorting
Before recording, we removed the dura above the opened left hemisphere of the cerebral cortex as well as that above the opened right hemisphere of the cerebellar cortex, fixed the mice in the apparatus and adjusted all manipulators under anesthetization with isoflurane. All recordings began at least 30 min after turning off the anesthesia. Neocortical recordings were made in M1 and S1 using 16-channel, single-shaft silicon probes with an inter-electrode tip distance of 100 μm (R = 1.5-2.5 MΩ, A1x16-5mm-100-177-A16, NeuroNexus Technologies, Ann Arbor, MI, USA). Each silicon probe was equipped with its reference, which was placed close to the recording site. The two probes shared the same ground, which was placed in the agar covering the recording sites. With these probes, we recorded the neocortical LFP signal throughout all cortical layers and single-unit neurons (62 in S1 and 55 in M1), as we previously described in Lindeman and colleagues[9]. These neurons were recorded at a comparable depth (S1 = 950 ± 363 μm, M1 = 979 ± 409 μm, p = 0.684; unpaired TTEST). At the same time, electrophysiological recordings were performed in the cerebellar crus regions in awake mice using quartz-coated platinum/tungsten electrodes (2–5 MΩ, outer diameter = 80 μm, Thomas Recording, Giessen, Germany) placed in a matrix 4 × 8 with an inter-electrode distance of 305 μm (Thomas Recording); the configuration of this matrix has been described in detail by Romano and colleagues[22]. We used a subset of 12 equidistant electrodes of this matrix for

cerebellar LFP recordings at the Purkinje cell layer (Fig. 1a). These electrodes were individually moved deeper until when a level of intense activity, typical of the Purkinje cell layer, was detected (range 250 to -1300 μm). The Purkinje cell single-unit recordings were done at a minimal depth of 250 μm. Both cerebellar and cerebral signals were acquired and stored using a single RZ2 multi-channel workstation (Tucker-Davis Technologies, Alachua, FL). Therefore, electrodes and silicon probes from different companies (i.e., NeuroNexus Technologies and Thomas Recording) were synchronized by acquiring them with the same device, software and timestamp (from Tucker-Davis Technologies). The raw electrophysiological signal was digitized at 25 kHz, using a 1–6000 Hz band-pass filter, 22x pre-amplified. All cerebellar spikes were detected offline using SpikeTrain (Neurasmus, Rotterdam, The Netherlands)[22–24]. A Purkinje cell signal was recognized by the appearance of the typical complex spikes and simple spikes. The criterion for a single-unit Purkinje cell recording was the minimal inter-spike interval of simple spikes of 3 ms and the minimum pause in simple spike firing of at least 8 ms after each complex spike. To avoid the firing rate change being due to the instability of the recording, we excluded recordings in which the amplitude or the width of more than three consecutive simple spikes exceeded three standard deviations above or below their average. Only those recordings during which the amplitude and the width of the spikes were constant over time were included in our study. When these criteria were satisfied, we considered them stable single-unit recordings of Purkinje cells. As the single-unit recordings were not always available throughout all recording locations (i.e., 16 channels in S1, 16 channels in M1 and 12 channels in the crus cerebellar region), we used the LFP (raw signal filtered with a low-pass filter of 80 Hz), which reflects both the synaptic input and spiking output of the neurons in the recorded location[64].

## Viral injections
Craniotomies were performed allowing access to the whisker part of the left primary somatosensory (wS1, relative to bregma: 3.5 mm mediolateral and -1.5 mm anteroposterior) and motor cortex (wM1, relative to bregma: 1.6 mm mediolateral and 1.7 mm anteroposterior)[9]. To express ChrimsonR in neocortical pyramidal neurons, 30 nl of AAV9-Syn-ChrimsonR-tdTomato viral vectors (Addgene, titer: $2 \times 10^{13}$ GC/ml) was injected in M1 and 30 nl in S1. Afterwards, a recording chamber was made and sealed with a silicon polymer (Kwik Cast, WPI, Sarasota, FL, USA). After 6 weeks of incubation, this chamber was used for electrophysiological recordings.

To express stGtACR2 in the cerebellar-targeted thalamic neurons we injected -90 nl of transneuronal anterograde AAV1-CMV-Cre-

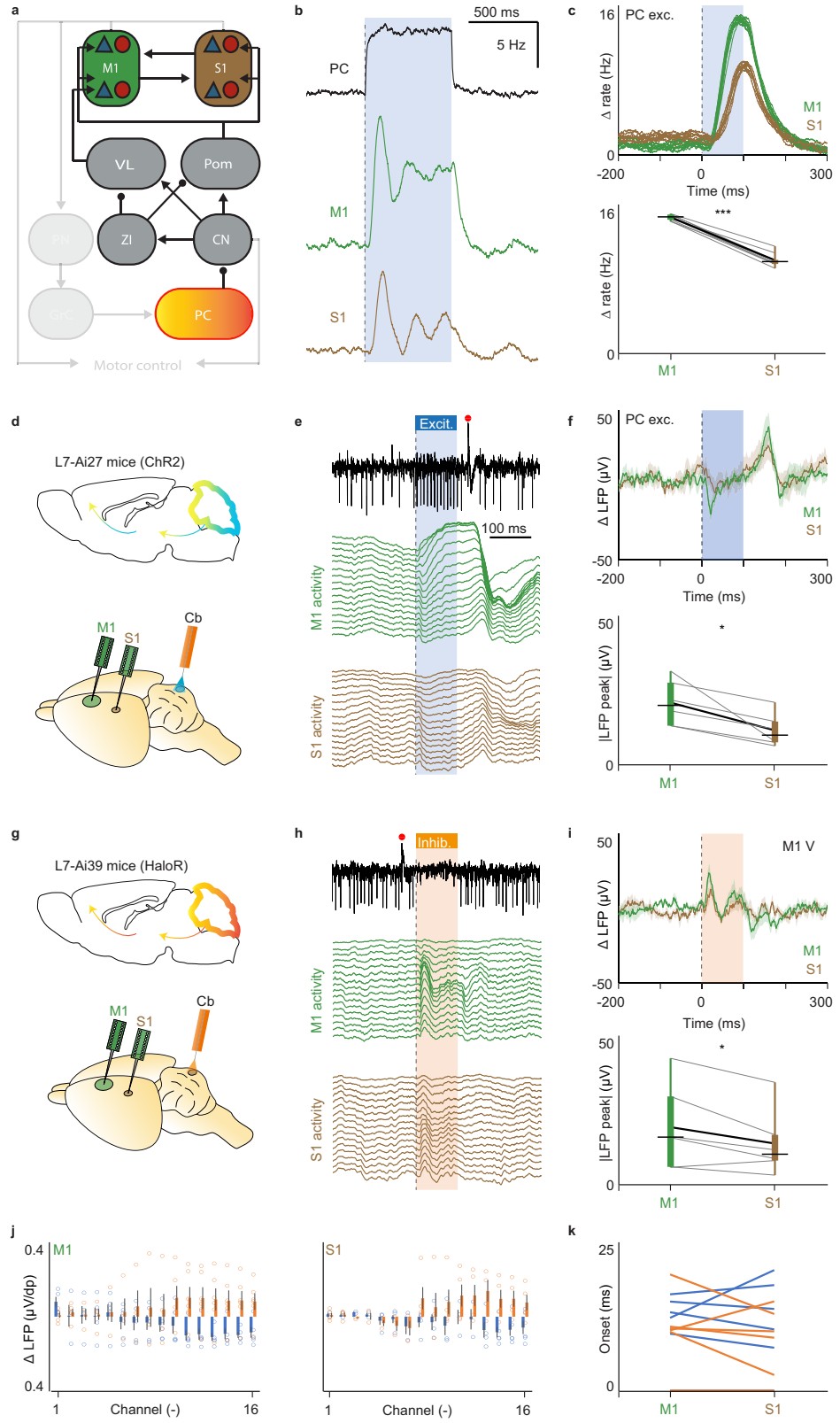

GFP (Addgene, titer: $2.7 \times 10^{13}$ GC/ml) in the cerebellar nuclei (relative to lambda: −2 mm mediolateral, −2.4 mm anteroposterior, −2.3 dorsoventral) and 30 nl of AAV1-hSyn1-SIO-stGtACR2-FusionRed (Addgene, titer: $2 \times 10^{13}$ GC/ml) in the thalamus (relative to bregma: −0.9 mm mediolateral, −1.3 mm anteroposterior, −3.5 dorsoventral). Likewise, the same viruses were used for the pontine stimulation experiments; here ~40 nl of transneuronal anterograde AAV1-CMV-

Cre-GFP was injected in M1 (relative to bregma: 1.6 mm mediolateral, 1.7 mm anteroposterior and -0.5 dorsoventral) and another ~40 nl in S1 (relative to bregma: 3.5 mm mediolateral, −1.5 mm anteroposterior and −0.5 dorsoventral), after which 30 nl of AAV1-hSyn1-SIO-stGtACR2-FusionRed was injected into the pons (relative to bregma: 0.5 mm, −4.2 mm anteroposterior and −4.7 dorsoventral). Only the mice in which the stimulation of the stGtACR2-expressing neurons

**Fig. 6 | Cerebellar output affects motor cortex more than sensory cortex.**
**a** Schematic of the modeled stimulation of PC resulting in the manipulation of the ascending pathway. **b** An example of the temporal development of different firing rates of M1 and S1 in the model as a result of PC stimulations. **c** Temporal evolution of the firing rates of M1 and S1 in the model as a result of ten different PC stimulations (top). The peaks of maximum responses in M1 and S1 are plotted for comparison (bottom, $p = 2.2612E-09$, Two sided paired TTEST). The bounds of the box indicate the interquartile range, the whiskers indicate the entire range, the black horizontal thin lines represent the medians, and the thicker black line represents the means. **d** Schematic of the experimental approach used to optogenetically excite PC activity while recording M1 and S1 activity. PC-specific genetic mouse models, expressing channel rhodopsin 2 (L7-Ai27, ChR2), were stimulated using blue light (470 nm). **e** Blue light induces a burst in simple spike (SS) activity and LFP responses throughout the layers of the primary motor (green, top) and sensory (brown, bottom) cerebral cortex for one exemplary ChR2 mouse. **f** Mean LFP responses for 6 ChR2 mice. The lines represent the mean (+/− SEM) LFP of one recording channel of M1 and one of S1, corresponding to the cortical layer V. The peaks of absolute maximum responses in M1 and S1 are plotted for comparison

(similar to c, but for the experimental data, $p = 0.011$; One sided paired TTEST). The bounds of the box indicate the interquartile range, the whiskers indicate the entire range, the black horizontal thin lines represent the medians, and the thicker black line represents the means. **g** Similar to panel d, but for the PC-specific genetic mouse model expressing halorhodopsin (L7-Ai39, HaloR). **h** In these mice orange light (595 nm) induces a pause in SS activity and alters the LFP responses in the primary motor (green, top) and sensory cortex (brown, below). **i** Same as panel f, but for PC inhibition ($p = 0.047$; Two sided paired TTEST), highlighting the more prominent impact on M1 than on S1. Lines represent the mean +/− SEM. **j** Bar plot of the significant LFP response onset (see Methods), for the 6 ChR2 (blue) and 6 HaloR (orange) mice. Each bar represents one of the 16 recording channels in M1 (left) and S1 (right). Note that bidirectional SS modulation resulted in bidirectional LFP modulation in both M1 and S1. Error bars indicate the standard deviation. **k** Comparison between the M1 and S1 LFP responses onsets for PC excitation ($p = 0.386$; One sided paired TTEST) and PC inhibition (p = 0.176; One sided paired TTEST). The comparison was made only for combination of channels in which M1 and S1 responses crossed the significant threshold (Z-score > 2; see Methods).

yielded an LFP response in the neocortex or cerebellum were used for further analysis. We used stGtACR2, because it has a reduced probability of generating antidromic spiking, compared to the classical GtACR2 (Fig. S6)[65,66].

## Optogenetic manipulation

For stimulations of the pons and thalamus we implanted a chronic optic cannula into the same stereotactic location of the stGtACR2 viral injection, and we used a blue light source ( ~ 460 nm, M470F3, Thorlabs, Newton, NJ, USA). In these optogenetics experiments, the "control" refers to the same mice with stGtACR2 expression, but with the LED light off, not to mice expressing a similar fluorescent protein without stgtacr2 and with the application of light. For specific manipulation of the Purkinje cells we used transgenic L7-Ai27D mice (expressing channelrhodopsin-2) and L7-Ai39D mice (expressing halo rhodopsin). LED photo-stimulation (wavelength = ~460 nm, M470F3, or wavelength = ~595 nm, M595F3, Thorlabs, Newton, NJ, USA) was given by a high-power light driver (DC2100, Thorlabs, Newton, NJ, USA) through an optic fiber (400 μm in diameter, Thorlabs, Newton, NJ, USA). The optic fiber was placed on the surface of the right cerebellar hemisphere on lateral crus 1 (−0.3 mm posterior to the horizontal fissure, 2 mm lateral to the vermal longitudinal sulcus). The neocortical stimulation was performed in the virally transfected non-transgenic mice placing two optic fibers, similar to the one used for cerebellar stimulation, on the surface of the left cerebral hemisphere. For both the neocortical and cerebellar stimulation, the intensities of the light stimulation varied from 1 to 10 mW, while the stimulation varied from 100 ms in 2 s intervals. For the pons and thalamic inhibition, we used a 200 ms pulse given at time intervals ranging from 0.5 to 3.5 s (i.e., 0.5, 1, 1.5, 2, 2.5, 3, 3.5 seconds) in a random sequence.

## Statistics and visualization

All data analysis was conducted on MATLAB versions r2023a and r2023b. Whisker movements were detected similarly to those[21,22], using our tracking tool[63]. Only whisker movement with at least 200 ms duration and no other movement in the 400 ms before were considered for the analysis. Per mouse, 233 ± 84 (mean ± standard deviation) whisker movements satisfied these criteria and were used for calculating the cross-correlation of Fig. 1. The pre-movement (200 ms before onset) and movement (200 ms after onset) were determined from the selected onsets. Resting periods (200 ms) were selected from the middle of non-movement windows of at least 400 ms. Cerebro-cerebellar correlation was calculated in these time windows using the MATLAB function "xcorr" which returns cross-correlograms between LFP signals over the same time window relative lead/lag. From each

cross-correlogram, we extracted the amplitude of the maximum peak and its lag (i.e., time displacement) of the cross-correlation MATLAB function (calculated from −100 ms to 100 ms), which we refer to as "phase relation". These values, relative to the correlation of individual pairs of channels, were averaged out for all movement epochs and across pairs of channels. To calculate the correlation expected by accidental synchronization, we computed the correlation of pairs of channel signals at random time points. This process was repeated 100 times, giving a distribution of results expected by chance. This was done for every combination of LFP channels, giving each combination a mean and standard deviation of random results. These were used to calculate the Z-score of the observed data. Such a Z-score was used to select the combination of cerebellar and cerebral signals with significant correlation (Z > 2). The details of the statistical tests are summarized in Table 1.

We calculated the onset of the optogenetic responses from the mean signal of the recorded LFP signals around stimulation. This signal was filtered with a low-pass filter of 80 Hz and subsequently differentiated. Eventually, we extracted the absolute maximum LFP peak (the furthest deviation from the mean of the signal before the stimulation) and its phase relation from the stimulus onset. To compare the observed data with the expected variability, a mean signal was calculated for each channel during non-stimulation. This signal was also filtered and differentiated, creating a baseline for non-stimulation. From this baseline, a mean and two standard deviations were used to identify significant LFP changes due to stimulation. When the average differentiated signal crossed this threshold, the response was considered significant and the timepoint in which it crossed was considered as the response onset. Such selection of significant responses was necessary to determine the response onset but was not used for the comparison of M1 and S1 responses. The difference between M1-cerebellar and S1-cerebellar correlations (Fig. 2) was calculated by extracting the peak of the strongest correlations for each movement onset and pre-movement epoch. In addition, we calculated the difference between the peaks of the maximum correlations and plotted them on the corresponding cerebellar matrix.

The Granger causality analysis was performed using the MATLAB function "gctest" around (in the 200 ms before and 200 ms after) the simultaneously recorded whisker movement onset. For the mutual information analysis, we used the MATLAB function "mi_cont_cont" for LFP-LFP, "mi_discrete_cont" for spikes-LFP and "MutualInformation" for spikes-spikes. These functions are available at https://nl.mathworks.com/matlabcentral/fileexchange/. Both spike and LFP data were pre-processed in that the spike data were converted into binary vectors and both were down-sampled at 1000 Hz. Data

including movement or electrical artefacts in the electrophysiological signal were excluded from this analysis. The current source density plot of Fig. S3a, b was done using kCSD-Matlab (RRID:SCR_016424). To assess whether the presence of double peaks in the cross-correlograms was a consistent pattern (Fig. 1c), we used the Matlab function "find-peaks". Using the setting "MinPeakWidth", 500, "MinPeakPromi-nence", 0.01, "MinPeakDistance", 500, on the signal sampled at 25 kHz. Such settings allowed us to detect the presence of double peaks also when they were not clearly separated from each other (Fig. S2f, top left). The cross correlation between the whisker movement and the LFP signal of M1, S1 and cerebellar cortex was calculated on the down-sampled LFP signal at 1 KHz (i.e., the sampling frequency of the tracked angle whisker position). For this analysis, the cross-correlograms could not be calculated in the Pre-movement stage, where the whisker angle position was near zero. We calculated them in the epoch where the biggest movement was observed (from 20 to 170 ms after the movement onset, Fig. S5a). The same whisker angle position of each movement was cross-correlated with the down-sampled LFP signal of each recording channel. For each cross-correlogram, we extracted the phase relation of its peak (indicating the lead or the lag, relative to the whisker movement) and averaged those values for each recorded brain area (i.e., M1, S1 and cerebellar cortex). Note that the LFP signal was set as the first value of the cross-correlation MATLAB function "xcorr", therefore, a negative peak or value of the phase relation indicates that changes in the LFP signal precede changes in the whisker movement and vice versa. Spike-spike correlation matrix on a trial-by-trial basis (Fig. S4d, e) was performed with an approach that we already used before[32]. In short: spike density functions were computed for all trials by convolving spike occurrences across 1 ms bins with an 8 ms Gaussian kernel. The data were not aligned to the baseline. The Pearson correlation coefficient r was calculated in bins of 10 ms, resulting in an $80 \times 80$ r-value matrix showing correlations for $-400$ to $400$ ms around the movement onset. The r-values of the two $20 \times 20$ matrices (one for the Pre-movement and another for the Movement stage) were considered for further analysis. We excluded the r-values on the diagonal and subtracted the remaining values above the diagonal from those below. Finally, we compared these differences in the Pre-movement versus the Movement stages.

The analysis on the rate of the whisking bout initiation (Figs. 4g and 5g) was performed creating a peri stimulus time histogram (PSTH) of the bout start events around the stimulus onset and quantifying the mean rate during (0 to 200 ms relative to the onset) and before (-200 to 0 relative to the onset) the stimulation. The power spectrum of the whisker movement (Figs. 4h and 5h) was calculated using continuous wavelet transform (CWT) in Matlab, which return the power of a given signal overtime. Then we selected all the stimuli for which the onset fell within an epoch of rhythmic whisking. Finally, we averaged the spectrograms in the 500 ms before and 500 ms after the stimulus onset. The duration of this time window is suitable to calculate whisker oscillation with a good resolution[22].

The schematic of (Fig. 1a) has been created using previously created elements (https://doi.org/10.5281/zenodo.3925903) and distributed under Creative Commons Attribution 4.0 International.

## Computational model of the cortico-cerebellar network

The computational model developed here uses parts of our previous work[9,39,40] as a basis. The model constitutes a large-scale network of neural-mass populations, which simulates the firing rates of multiple interconnected cortical, thalamic and cerebellar populations, in a way that is consistent with existing electrophysiological and neuroanatomical evidence. For the neocortex, each cortical area is constituted by two cortical layers (or more generally, laminar modules) describing, respectively, the dynamics of superficial and deep layers. A laminar module contains one excitatory and one inhibitory population, and the dynamics of their respective firing rates $r_E(t)$ and

$r_I(t)$ are given by

$$\tau_E \frac{dr_E(t)}{dt} = -r_E(t) + F(I_E) + \sqrt{\tau_E}\sigma\,\xi(t) \tag{1}$$

$$\tau_I \frac{dr_I(t)}{dt} = -r_I(t) + F(I_I) + \sqrt{\tau_I}\sigma\,\xi(t) \tag{2}$$

Here, parameters $\tau_E$, $\tau_I$ determine, respectively, the time scales for the excitatory and inhibitory populations. The terms $\xi_E(t)$, $\xi_I(t)$ are Gaussian white noise variables of zero mean and standard deviation *one*. Following previous work, we choose $\tau_E = 6$ ms, $\tau_I = 15$ ms and $\sigma = 0.3$ for superficial layers, and $\tau_E = 48$ ms, $\tau_I = 120$ ms and $\sigma = 0.45$ for deep layers. $F(x) = x/(1 - \exp(-\beta x))$ constitutes the transfer function of each population, which transforms the incoming input currents into cell-averaged firing rates, with $\beta = 1$ being the firing onset sensitivity parameter. The argument of the transfer function is the incoming current for each population, and it contains a background term, a local term and a long-range term. The background term is a default constant current only received by excitatory neurons in S1 and M1, and it is $I_{bg} = 4$ for superficial excitatory neurons and $I_{bg} = 1$ for deep excitatory neurons. The local term involves the input coming from neurons within the area, and it is given by

$$I_{\text{local}}^E = 1.5\,r_E - 3.25\,r_I + I_{\text{interlaminar}}^E \tag{3}$$

$$I_{\text{local}}^I = 3.5\,r_E - 2.5\,r_I + I_{\text{interlaminar}}^I \tag{4}$$

Here, the numbers denote the strengths of the synaptic projections considered. The interlaminar terms are contributions from a different layer than the one the population is in. The only interlaminar projections are from superficial excitatory to deep excitatory neurons, with a synaptic strength of 1, and from deep excitatory to superficial inhibitory neurons, with a synaptic strength of 0.75[40]. Finally, the long-range term includes currents coming from other neocortical or subcortical areas. These currents follow the general form $J_{ab}r_b$, (with $J_{ab}$ being the synaptic strength from area 'b' to area 'a') and therefore we will specify only the synaptic coupling strengths to characterize them. As anatomical evidence suggests[25], we consider excitatory projections from superficial S1 neurons to both superficial (strength 0.52) and deep (0.25) excitatory M1 neurons, and from deep S1 neurons to superficial (0.25) and deep (0.75) excitatory M1 neurons. In the opposite direction, we consider excitatory projections from superficial M1 neurons to both superficial (0.5) and deep (1) S1 excitatory neurons, and from deep M1 neurons to deep (1) S1 excitatory neurons.

The dynamics of the firing rate of the thalamic nuclei (PN, VL, Pom), zona incerta (ZI), and cerebellar populations, granule cells, Purkinje cells and cerebellar nuclei neurons, are each described by equations of the type

$$\tau \frac{dr(t)}{dt} = -r(t) + f(I) \tag{5}$$

Here, $\tau = 6$ ms is the characteristic time constant and the transfer function is $f(x) = Ax/(1 - \exp(-\beta(x - \theta)))$, with A being the gain of the population, $\beta = 15$ the onset sensitivity and $\theta$ an input background (which may be positive or negative depending on the population). This function provides a good approximation of the input-output properties of spiking neurons[67], and is therefore preferred here to other, more abstract options like ReLU or sigmoid functions. The gain parameter A takes the values 10, 1, 1, 5, 0.2, 1 and 1 for Purkinje cells, cerebellar nuclei neurons, ZI, VL, Pom, PN and Granule cells, respectively. Likewise, the background input $\theta$ takes the following values for the same populations: 0.1, 21, -12, 0, 30, 0, and 0. Besides this

background input, cerebellar nuclei neurons and ZI receive inhibitory projections (both with synaptic strength 1) from Purkinje cells and cerebellar nuclei neurons, respectively. Pom receives projections from the cerebellar nuclei (strength 0.2) and ZI (strength -0.5). VL also receives projections from the cerebellar nuclei neurons (strength 1) and ZI (strength -3). Projections from Pom reach all neural populations in S1, with strengths of 0.3 (for excitatory populations) and 0.1 (for inhibitory ones), as well as all populations in M1, with strengths of 0.33 (for excitatory ones) and 0.5 (for inhibitory ones). VL also projects to excitatory cells in M1 deep layers with a strength of 0.6. The above parameter values for population gain, background and synaptic strength do not necessarily correspond to realistic estimations (since such experimental data are not available for the entire cerebello-thalamo-cortical network), but have been obtained after a careful computational exploration to place the overall circuit in a working point to deliver proper modeling predictions. These values must be understood as rough estimates, and while automatic optimization procedures could lead to better parameter settings, our results turned out to show that these approximations work sufficiently well to replicate the experimental findings. Parameters corresponding to cortico-cortical interactions and cerebello-thalamo-cortical pathways have already shown to be useful in replicating previous data sets[9,40].

For the analysis of correlations between populations (Fig. 3) we consider the presence of Ornstein-Uhlenbeck (OU) noise, whose dynamics are given by:

$$\tau_x \frac{dx(t)}{dt} = -x(t) + \sqrt{\tau_x}\, \sigma_x \xi(t) \qquad (6)$$

We consider (i) one OU process arriving at the Purkinje cell (with $\tau_x = 80$ ms and $\sigma_x = 0.5$), (ii) a shared OU process arriving at both Purkinje cell and S1 ($\tau_x = 80$ ms and $\sigma_x = 2$), and (iii) an OU process arriving at deep layers of M1 ($\tau_x = 200$ ms and $\sigma_x = 100$) only during the preparatory movement phase. The first of these OU signals is meant to introduce a natural level of variability in the activity of Purkinje cells, the second signal corresponds to a joint sensory afferent arriving at both Purkinje cells and S1 (which is based on neuroanatomical evidence and is partially responsible, in this model, for their high coherence level), and the third signal correspond to the 'ramping' preparatory signals found in M1 during the pre-movement phase (although its dynamics corresponds to a low-pass filtered Gaussian noise, and it disappears as soon as the model enters the movement phase for simplicity).

Optogenetic pulses (Figs. 4 and 5) were simulated by a current of 0.6 during a time window of 100 ms or 500 ms. To mimic the depth of the recording electrodes for S1 and M1 in experiments, we estimate the LFP signal in the model by a weighted average of the activity of excitatory superficial and deep layers, with a superficial to deep ratio of 20:80 for both S1 and M1. This ratio reflects the larger contribution of layer 5/6 pyramidal neurons in LFP signals, as these neurons are typically larger in size and have longer apical dendrites than superficial pyramidal cells.

### Reporting summary
Further information on research design is available in the Nature Portfolio Reporting Summary linked to this article.

## Data availability
All data are available from the Lead Contact upon request. Part of the data used for this paper is also available for reuse on a public repository "Simple and complex spikes of cerebellar Purkinje cells and simultaneously tracked whisker voluntary movement"; Romano, V., Zhai, P., & De Zeeuw, C. I. (2022); EBRAINS https://doi.org/10.25493/42NK-RYF. The raw data in this study are available from the corresponding author (V. Romano) upon request, owing to the size and complexity of the datasets. Source data are provided with this paper https://doi.org/10.6084/m9.figshare.29457575.

## Code availability
The custom code complementing BWTT whisker tracking can be obtained via https://github.com/elifesciences-publications/BWTT_PP our whisker tracker can be obtained via (https://gitlab.com/c7859/neurocomputing-lab/whisker/whiskeras-2.0. The code to simulate the computational model may be accessed via this link: https://modeldb.science/2019877.

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

## Acknowledgements

We thank M. Rutteman and E.D. Haasdijk (Department of Neuroscience, Erasmus MC) for their technical assistance. This research was funded by the Dutch Research Council (NWO): VI.Veni.222.329; (VR). Support was also provided by ERC-Adv, ERC-PoC, EU-LISTEN, Medical NeuroDelta, NWO-ALW, ZonMw, INTENSE NWO-LSH (C.I.D.Z.), NWA–ORC grant NWA.1292.19.298, UvA/ABC Project grant 2021-1060, the EU Horizon Europe Program under the specific grant agreement 101137289 "Virtual Brain Twin Project" (J.F.M.) and DBI2 NWO Gravitation Program (C.I.D.Z. and S.B.).

## Author contributions

C.I.D.Z., J.F.M., M.v.D. and V.R. designed the experiments. N.v.W. and V.R. performed the experiments. J.F.M. performed the simulations of the computational model. M.v.D., B.B., S.B. and V.R. analyzed the data and contributed to data visualization. C.I.D.Z., V.R., M.v.D. and wrote and revised the article. V.R. and M.v.D. contributed equally to the study.

## Competing interests

The authors declare no competing interests.
