## [Transparent Peer Review file · Nature Communications]

Stage-dependent cerebrocerebellar communication during sensorimotor processing

Corresponding Author: Dr Vincenzo Romano

Version 0:

Reviewer comments:

Reviewer #1

(Remarks to the Author)

Summary:

This study investigates the differential contributions of the cerebellum to motor planning and execution during spontaneous whisker movements in mice. By recording local field potentials from the motor cortex, somatosensory cortex, and cerebellar crus, the authors demonstrate that, during movement preparation, cerebellar activity correlates with both cortical areas but exhibits a temporal lag. In contrast, during movement execution, the phase relationship of this correlation reverses, and the cerebellum's coupling with the somatosensory cortex strengthens. The authors interpret these findings as evidence that the cerebellum receives an efference copy of the motor plan during preparation and, during execution, shifts to comparing incoming sensory feedback with its internal predictive model, issuing corrective signals when discrepancies arise.

The study employs a well-designed electrophysiological approach, simultaneously recording from three key regions and leveraging a behaviour - whisker movement - that is highly appropriate for probing the central hypothesis. The data and correlogram-based analyses are promising and address a timely question in neuroscience: how complex functions emerge from the dynamic interactions between multiple brain regions. However, the authors may overinterpret their results by equating correlation with neural representation. Incorporating additional behavioural analyses to directly relate electrophysiological signals to movement would provide stronger empirical support for their interpretations and further enhance the study's impact.

Major concerns:

1. As mentioned, The authors' interpretation of correlogram lags as indicative of distinct movement representations may be premature without linking neural signals to specific whisker-movement features. A more rigorous approach would involve explicitly relating local field potentials to whisking behaviour. One possibility is to cross-correlate the whisker-movement signal with each region's LFPs in the same way that the authors presently compare signals across brain areas. If M1 and the cerebellum truly encode a motor plan in the preparatory phase, one would expect a predictive lag (potentially longer in the cerebellum) relative to the whisker signal. Likewise, if S1 encodes a retrospective sensory representation during execution, one might anticipate an inverted cross-correlation pattern aligned with whisker movements.

An alternative and perhaps cleaner approach would be to employ autoregressive-like modelling, using a temporally shifted version of the whisker-movement signal to predict the LFPs (e.g. having 60 regressors from $t-30$ to $t+30$) and then removing all nonrelevant temporal regressors outside the target window. This method could more precisely decompose the LFP into past (e.g. $t-30 - t-10$), current (e.g. $t-10 - t+10$), and future (e.g. $t+10 - t+30$) correlates of movement, and thereby clarify how each area contributes to representing plans vs past movements. Re-examining inter-regional cross-correlations with this filtered signal would provide a stronger basis for the claim that M1, S1, and the cerebellum perform distinct computational roles in whisker control.

2. The rationale for focusing solely on the retraction phase of the movement, rather than analysing the entire whisking cycle, is unclear. Given that sensory feedback-driven corrections are likely to be most pronounced in the later stages of the movement, a more comprehensive analysis that includes both protraction and retraction periods could provide a fuller

picture of cerebellar involvement in movement control. Could the authors please clarify their choice.

3. In the Introduction, the authors suggest that the cerebellum's heightened correlation with S1 - alongside its driving role during movement execution - underlies the corrective aspects of motor control. Since this cerebellum-S1 interaction forms a key component of the authors' conceptual model, it would be instructive to test whether the sensorimotor copy is indeed crucial for stabilising ongoing movements. One possible approach could be to test whether the cerebellum contributes to the stability of oscillatory whisking patterns, particularly in prolonged whisking bouts. For instance, does the cerebellar efference copy help maintain a consistent whisking frequency, and would disrupting its input to M1 impair this rhythmic precision? Ontogenetically silencing cerebellar input to M1 (The "Role of the ascending pathway" experiment) provides a direct test, and I would expect this manipulation to broaden the peak at ~15 Hz in the whisking frequency spectrum. I am assuming here that there were enough spontaneous whisker movements during the inhibition periods to run such an analysis (which might be not the case).

4. On a related, but more general note, the optogenetics experiments would benefit from an exploration of behavioural data. How did each manipulation influence behaviour, and can these effects be linked to the representations required for movement control as outlined in the introduction? For example, the authors conclude that the descending pathway provides a copy of motor commands generated in the cerebral cortex to the cerebellum, which may facilitate the movement preparation. Does blocking this pathway result in the cerebellum not representing the future component of movement, and does this mean that on average movements were generated less frequently within the stimulation periods?

5. The manuscript lacks a clear motivation for the use of the computational model described in the text. While the authors employ the model to demonstrate that silencing relay regions to and from M1 influences neural correlation lags in a manner consistent with predictions, it remains unclear why this particular model is well-suited for addressing these questions. Specifically, the authors propose in the results section that the observed lags arise from the distinct computational roles of different brain regions at various stages of movement control. However, the model primarily captures connectivity and some biological details of the recorded areas, without being explicitly trained to generate movements or endowed with inductive biases that would naturally give rise to the computations hypothesised in the introduction and stated in the results section. A stronger justification for the model's relevance would improve the manuscript. Specifically, how does reproducing timing and delays in a handcrafted architecture provide new mechanistic insights? This concern is particularly relevant given the absence of behavioural readouts in the model. Additionally, the sections about the ascending and descending pathways would strongly benefit from detailed explanations of the predictions derived from the model.

6. Related to the point above it would be useful to discuss in some detail both in the intro (e.g. in lines 66-68, lines 96-98) and discussion (see below) how does the current work contrast with recent computational models of cortico-cerebellar interactions (Pemberton et al. Nature Comms 2024: <https://www.nature.com/articles/s41467-024-55315-6>, but also Boven et al. Nature Comms 2023). Those models capture observations of cortico-cerebellar interactions, while making several predictions. These models can be interpreted as a forward/predictive model and explains many observations, including motor preparation. Note that the Pemberton 2024 study is perhaps the most relevant as its closer to your model. For example, when you write "As the feedforward model requires feedback during the actual execution of the movement to optimize its future predictions.." in the discussion, it would be relevant to discuss this model. Moreover, "to the cerebral cortex to improve subsequent motor commands" is very consistent with the model proposed by Pemberton et al. In general, the discussion is quite short, so it would benefit of this and other additions. But also the introduction.

Minor:

1. "The argument of the transfer function is the incoming current for each population, and it contains a background term, a local term, and a long-range term." Could the authors clarify why this specific function was chosen instead of more commonly used activation functions such as sigmoid, ReLU, or tanh?

2. What is the rationale for selecting a 20:80 proportion for LFP computation?

3. The input to the model and its role in generating movement are not entirely clear. Does the model produce movement explicitly, or does it merely simulate neural activity related to movement?

4. Could the authors elaborate on why laminar layer modules are critical for the movement control model being explored? How do they contribute to the conceptual framework?

5. It would be helpful to mention the optogenetic stimulation periods in the results section where relevant.

6. All t-tests should be reported with degrees of freedom, t-values, and effect sizes.

7. Significance markers in figures should indicate p-value levels explicitly.

8. In Figure 1c-e, is the data from a single animal, while Figure 1f represents the group-level data? Please specify this in the text.

9. Figure 2a appears redundant.

10. In Figure 2b-c, what exactly does the C-C coefficient represent? Is it the average cross-correlation in a 200 ms window across all pairs, computed per animal? Or is it the peak value?

11. For Figure 1, how many trials contribute to each cross-correlation estimate per animal? Additionally, what is the standard deviation?

(Remarks on code availability)

Reviewer #2

(Remarks to the Author)

In this manuscript, Romano et al. studied the dynamic information flow between M1/S1 and the cerebellar cortex right before and after spontaneous whisker movements. Paired LFP recordings, optogenetic manipulations, computational modelling and spike recording techniques were used in order to support their conclusion. I have some concerns about the methodology, some of the results and interpretation, which I listed below.

Major concerns

1. Analysis of paired LFP recordings can suggest information flow between brain regions. However, I think relying mainly on LFP analysis is not enough. LFP signals integrate neural activities across a broad spatial scale and may introduce confounds when attempting to infer communications between brain regions. The authors should provide additional validation, possibly through spike analysis (mutual information, entropy...) and Granger causality.

Waveform of LFP signals depends on recording locations. While it's nice to see results from 16 channels, it's also obvious that they are quite different (e.g., Figs 1a, 6j). Instead of simply averaging across channels, the authors may want to have more analysis on individual channels. For example, signals from the 16 linear channels in M1 and S1 may be biased towards different layers. Since the information flow among different cortical layers have been previously well characterized, this analysis may be used to validate the methodology (i.e. determine information flow using paired LFP recordings).

2. Some of the results may not support the final conclusion. i). A major conclusion of the manuscript is that during movement, LFP signals in Cb precede those in M1 (fig. 1f). However, in fig. 4f (top right, orange color), all results exhibit the opposite. ii). Statistics in fig. 2f and 2g may need double check. $P = 0.0496$ seems very weak. Is 3/55 and 7/62 enough to suggest stronger Purkinje cell activity correlation with S1? Are the layer distribution of these units in S1 and M1 comparable? iii). In line 270, it's stated "the polarity of the effect following this inhibition was opposite to that following excitation in both M1 and S1". However, in fig. 6j, results from channels 1-5 don't support this claim. This point may be worth discussion.

3. The rationale behind the computational modelling is not very clear. The authors should explain how this modelling advances our understanding of the system beyond what is achieved through experimental observations. From reading the current manuscript, I was left unclear of what extra insight the modelling provides.

4. Looking into details of the modelling part, I have more questions. i). Can authors provide more details about the "preparatory ramping signals"? It looks to me that this setting is the major factor that results in different lagging values between pre-movement and movement. ii). If this is true, the authors may want to test different ramping parameters and check their effects on simulation results. iii). "an OU process arriving at deep layers of M1". Does this setting actually cause the higher cerebellar correlation with S1? After all, $\sigma_x = 100$ will make it VERY noisy. iiiii). Many of the parameters were manually determined. What's the rationale behind these parameter settings? iiiiii). The authors later uploaded original codes for their modelling, which I really appreciate. In the codes, authors separate "Pre-movement" and "Movement" into different "conditions". However, actual movement should be a continuous mixture of "Pre-movement" and "Movement". I think this may be the place where computational modelling can kick in and provide insights.

5. GtACR is known to induce antidromic spikes (PMID: 30297821). Considering the local connections in PN and TH, the authors may want to verify that light stimulation does not cause activation.

6. The authors suggest that the phase difference is consistent with the cerebellum receiving a copy of the cerebral commands during movement preparation, and the cerebellum provides feedback to the cortex during movement execution. While this is an interesting hypothesis, the forward model also suggests that the cerebral cortex need to send continuous motor command copies to the cerebellum during motor execution, in order to generate predicted sensory feedback for fast real-time motor adjustments. The representative results in fig. 1c on the right did exhibit double peaks both before and after movement onset. Is this pattern consistent?

In fact, it's also plausible that during the motor preparation, the cerebellum provides feedback to the cortex. Theoretically this can help with movement planning. I think it's OK to mention these and other hypothesis in the discussion part of the manuscript. However, if claiming these points in the other parts of the manuscript, more work will be needed.

Minor concerns:

1. What's the correlation coefficient for data in Figs 4 and 5?
2. Spontaneous whisker movement is likely not a goal-directed voluntary movement. Can the observations here be generalized to other types of movement? Some discussions of this topic may be helpful.
3. The volume of viral injections seem to be all wrong in the methods part. It's hard to imagine that 30-40 ul of virus can be infused to a single brain region...
4. The manuscript lacks a detailed description of how the two recording systems were synchronized. Precise timing is critical in this study.
5. In figure 1, it should be fig. 1f instead of 1F.
6. What's the titer of AAVs used in this study?

(Remarks on code availability)

Reviewer #3

(Remarks to the Author)

(Remarks on code availability)

Version 1:

Reviewer comments:

Reviewer #1

(Remarks to the Author)

We thank the reviewers for addressing our comments in detail. We are happy with the revised manuscript.

(Remarks on code availability)

Reviewer #2

(Remarks to the Author)

The revised version addressed most of my concerns. Here I listed a few points which hopefully are helpful for the authors.

Firstly, the "control" group in Figure 4f and 5f need to be defined clearly. I initially wrote a long paragraph explaining why data in Figure 4f still doesn't fit with the conclusion that Cb signals precede M1 during movement. Later I deleted it, because I realized that the "control" condition in this study is not fluorescent protein expression plus light on, but stGtACR2 expression plus light off (is that right?). And now the response in rebuttal letter all makes sense to me. Please clearly define the "control" condition in optogenetics experiments, so that some readers don't need to experience the confusion I do. Although I still think fluorescent protein control is better for optogenetics experiments (as the authors suggested, repeated manipulation have after-effects), this is more of a suggestion for future experiments, rather than a suggestion for revision, since it doesn't affect the conclusion of this study.

Secondly, the authors need to double check their statistical methods. Currently all statistical analysis is based on t-test. However, some data obviously don't follow a normal distribution (e.g., Figure 2h).

Lastly, the authors need to be more careful with manuscript writing. I listed some errors that I noticed during reading. But please proofread the manuscript on your side.

- 1). Figure legend for Figure 2h is completely missing;
- 2). Line 30, "align actual with". This expression seems not right;
- 3). Line 166, it should be "Figure 2f" instead of "Figure 2e";
- 4). Line 430, it should be "fine-tuning" instead of "finetuning";
- 5). Line 483, "similar to 13". This citation format seems not right;
- 6). Line 489, "see 23". Same citation format issue;
- 7). Line 507, it should be "unit" instead of "unite";

8). In figures, "GtACR2" should be replaced with "stGtACR2".

(Remarks on code availability)

Reviewer #3

(Remarks to the Author)

(Remarks on code availability)

REVIEWER COMMENTS

Reviewer #1

Summary:

This study investigates the differential contributions of the cerebellum to motor planning and execution during spontaneous whisker movements in mice. By recording local field potentials from the motor cortex, somatosensory cortex, and cerebellar crus, the authors demonstrate that, during movement preparation, cerebellar activity correlates with both cortical areas but exhibits a temporal lag. In contrast, during movement execution, the phase relationship of this correlation reverses, and the cerebellum's coupling with the somatosensory cortex strengthens. The authors interpret these findings as evidence that the cerebellum receives an efference copy of the motor plan during preparation and, during execution, shifts to comparing incoming sensory feedback with its internal predictive model, issuing corrective signals when discrepancies arise.

The study employs a well-designed electrophysiological approach, simultaneously recording from three key regions and leveraging a behaviour - whisker movement - that is highly appropriate for probing the central hypothesis. The data and correlogram-based analyses are promising and address a timely question in neuroscience: how complex functions emerge from the dynamic interactions between multiple brain regions. However, the authors may overinterpret their results by equating correlation with neural representation. Incorporating additional behavioural analyses to directly relate electrophysiological signals to movement would provide stronger empirical support for their interpretations and further enhance the study's impact.

We thank the Reviewer for highlighting the timeliness of our study and for the constructive comments to further improve the manuscript.

Major concerns:

1. As mentioned, The authors' interpretation of correlogram lags as indicative of distinct movement representations may be premature without linking neural signals to specific whisker-movement features. A more rigorous approach would involve explicitly relating local field potentials to whisking behaviour. One possibility is to cross-correlate the whisker-movement signal with each region's LFPs in the same way that the authors presently compare signals across brain areas. If M1 and the cerebellum truly encode a motor plan in the preparatory phase, one would expect a predictive lag (potentially longer in the cerebellum) relative to the whisker signal. Likewise, if S1 encodes a retrospective sensory representation during execution, one might anticipate an inverted cross-correlation pattern aligned with whisker movements.

An alternative and perhaps cleaner approach would be to employ autoregressive-like modelling, using a temporally shifted version of the whisker-movement signal to predict the LFPs (e.g. having 60 regressors from $t-30$ to $t+30$) and then removing all nonrelevant temporal regressors outside the target window. This method could more precisely decompose the LFP into past (e.g. $t-30 - t-10$), current (e.g. $t-10 - t+10$), and future (e.g. $t+10 - t+30$) correlates of movement, and thereby clarify how each area contributes to representing plans vs past movements. Re-examining inter-regional cross-correlations with this filtered signal would provide a stronger basis for the claim that M1, S1, and the cerebellum perform distinct computational roles in whisker control.

We thank the Reviewer for suggesting additional forms of analyses to strengthen our conclusions. We opted for the suggestion to directly relate the whisker movement signal with the LFPs of each region, similarly to what we did to compare signals across brain areas. We created a new figure (Figure S5) in which we indeed show that the LFPs of M1 and cerebellum on average precede the whisker movement, whereas those of S1 follow it. The differences between S1 and cerebellar LFPs were highly significant ($p = 0.002$), which fits well with the possibility mentioned by the Reviewer.

2. The rationale for focusing solely on the retraction phase of the movement, rather than analysing the entire whisking cycle, is unclear. Given that sensory feedback-driven corrections are likely to be most pronounced in the later stages of the movement, a more comprehensive analysis that includes both protraction and retraction periods could provide a fuller picture of cerebellar involvement in movement control. Could the authors please clarify their choice.

We thank the Reviewer for bringing up this important point. It shows that we were not sufficiently clear in our initial description. The time window we consider as “Movement” can in fact include both protraction and retraction. In the new version, we now state explicitly at the beginning of the Results section that both types of movements are included in the analyses.

3. In the Introduction, the authors suggest that the cerebellum’s heightened correlation with S1 - alongside its driving role during movement execution - underlies the corrective aspects of motor control. Since this cerebellum-S1 interaction forms a key component of the authors’ conceptual model, it would be instructive to test whether the sensorimotor copy is indeed crucial for stabilising ongoing movements. One possible approach could be to test whether the cerebellum contributes to the stability of oscillatory whisking patterns, particularly in prolonged whisking bouts. For instance, does the cerebellar efference copy help maintain a consistent whisking frequency, and would disrupting its input to M1 impair this rhythmic precision? Ontogenetically silencing cerebellar input to M1 (The “Role of the ascending pathway” experiment) provides a direct test, and I would expect this manipulation to broaden the peak at ~15 Hz in the whisking frequency spectrum. I am assuming here that there were enough spontaneous whisker movements during the inhibition periods to run such an analysis (which might be not the case).

The Reviewer is right. To test whether the cerebellar efference copy helps to maintain a consistent whisking frequency, we compared the whisking frequency spectrum before and after optogenetically silencing cerebellar input to M1. We found indeed less consistent rhythmic whisking, which was particularly prominent at the lower frequency range (3-10 Hz). This is in line with our previous finding that cerebellar stimulation is particularly effective in generating rhythmic whisking at 8 Hz (Bauer et al., 2022). We have now added new panels to Figures 4 and 5 to show these results and we highlight this point in the Results.

4. On a related, but more general note, the optogenetics experiments would benefit from an exploration of behavioural data. How did each manipulation influence behaviour, and can these effects be linked to the representations required for movement control as outlined in the introduction? For example, the authors conclude that the descending pathway provides a copy of motor commands generated in the cerebral cortex to the cerebellum, which may facilitate the movement preparation. Does blocking this pathway result in the cerebellum not representing the future component of movement, and does this mean that on average movements were generated less frequently within the stimulation periods?

This is another good point. To probe the influence of each manipulation on behaviour, we tested the impact of inhibition of both the descending and the ascending pathways on the consistency of rhythmic whisking (as we described above for cerebellar input to M1) as well as on the frequency of self-generated movement. We now show in the new figure panels that each pathway has an impact only during the expected stage (descending on initiation and ascending on rhythmicity). The direction of the impact of stimulation is in line with the study by Kleinfeld and colleagues (Deschênes et al., 2016 Neuron), who showed that whisking is driven by an inhibitory circuit. We address these points in the Results section.

5. The manuscript lacks a clear motivation for the use of the computational model described in the text. While the authors employ the model to demonstrate that silencing relay regions to and from M1 influences neural correlation lags in a manner consistent with predictions, it remains unclear why this particular model is well-suited for addressing these questions. Specifically, the authors propose in the results section that the observed lags arise from the distinct computational roles of different brain regions at various stages of movement control. However, the model primarily captures connectivity and some biological details of the recorded areas, without being explicitly trained to generate movements or endowed with inductive biases that would naturally give rise to the computations hypothesised in the introduction and stated in the results section. A stronger justification for the model's relevance would improve the manuscript. Specifically, how does reproducing timing and delays in a handcrafted architecture provide new mechanistic insights? This concern is particularly relevant given the absence of behavioural readouts in the model. Additionally, the sections about the ascending and descending pathways would strongly benefit from detailed explanations of the predictions derived from the model.

We appreciate that the Reviewer would like us to highlight better what the contributions of the model are and why our model is suitable. We would like to start off by noting that the core architecture of our model is based on the main existing neuroanatomical pathways and proven cortical dynamical models. Thus, replicating timing and delays in the architecture allows us to validate that the fundamental, data-driven network proposed by the model is in principle sufficient to explain the emergence of the experimentally observed dynamics. This is highly relevant, as several side-loops of the comprehensive connectome of the real brain were left out in our model architecture of cerebrocerebellar interactions. For example, we did not integrate the impact of the striatum or the role of the mesodiencephalic junction. Both side-loops are superimposed onto the main circuitry that we modelled, but they don't seem to play a critical role in the dynamics. For this the main descending route through the pontine nuclei and the main ascending route through the thalamus, together forming the closed cerebrocerebellar loop, appear to be sufficient. Although we agree that this was not sufficiently highlighted in the main text of the previous version of our manuscript, we frankly do think this is an important statement. Moreover, the main conclusions from our computational analyses also provide direct and specific mechanistic insight in that we showed that the lead of cortical dynamics over cerebellar dynamics can be explained by preparatory motor signals transmitted via top-down pathways, that such lead can be inverted by simply controlling the strength of preparatory signals in M1, without further changes in cerebellar cortex, S1 or thalamic nuclei, and that the higher cerebellum-S1 correlations and stronger effect of cerebellum on M1 can be explained by (i) a shared sensory input between cerebellum and S1, coupled with (ii) an overall stronger cerebellum -> M1 pathway. Besides this, the model also delivered testable (and tested) predictions for the experiments. For example, inactivating the pontine network should (and did) impair cortico-cerebellar communication during pre-movement, but not for movement; inactivating the VL thalamus should (and did) impair the cerebello-

cortical communication during movement, but not so much during the pre-movement phase; and a strong cerebellar drive should (and did) trigger a higher response in M1 than that in S1.

In addition, we are confident that our model is well-suited for our purposes, because (i) our data is relatively rich in electrophysiological features and focused on a naturalistic but simple behavior (self-initiated whisker movements) rather than on a complex, well-structured behavioral task, and (ii) the phenomenology that we attempt to explain here involves multiple large-scale structure and pathways across the brain. Regarding point (i), dynamical system models, and neural-mass models as in our particular case, are typically good choices to replicate and explain electrophysiological features such as cross-regional LFP correlations and lesioning/inactivation effects, as we have shown for example also in our previous modeling work on cerebello-cortical networks (Lindeman et al. PNAS 2021). Dynamical system models, on the other hand, indeed tend to perform relatively poorly when the specific movement patterns have to be captured or accurately described. Computational models based on RNNs trained on specific behavioral tasks, such as the studies of Boven et al. 2023 and Pemberton 2024 suggested by the Reviewer, excel in these aspects and are therefore the recommended choice when the focus is on complex, yet specific, behavioral tasks (such as line-drawing or digit-drawing tasks in the Pemberton study). However, the level of biophysical detail for RNN-based models (and therefore their capacity to replicate electrophysiological features) is often not as high as in the case of dynamical models like ours. For our particular goal it is therefore more convenient to make use of neural-mass models and focus on replicating electrophysiological findings. Regarding point (ii), neural-mass models are arguably the best choice to describe the neural dynamics of large-scale networks, such as the cerebello-thalamo-cortical circuits addressed here. This is a major reason for them being the default choice for cross-regional modeling (see Breakspear Nat. Neurosci. 2017 and Hancock et al. Nat. Rev. Neurosci. 2025). The modeling framework based on RNNs is still not at the same level in terms of incorporating neuroanatomical structures, despite its excellent handling of high-dimensional local neural dynamics. For example, the works of Boven and Pemberton referred to above are able to model the high-dimensional dynamics of a single cortical network and a single cerebellar network, while our present model describes the low-dimensional dynamics of about 15 neural populations. Given that our focus is to explain low-dimensional features (cross-population correlations, temporal lags, etc) across multiple areas, this provides another reason to use neural-mass networks in our manuscript. Note, however, that future work should ideally aim to merge these two modeling currents to capture the best of both frameworks (see also statement in van Holk and Mejias, 2024).

In the revised version, we have now included more extensive details on the motivation and suitability of our model, its relationship with existing models such as RNN-based ones mentioned above, as well as a clear description of the insights gained by each of our computational analyses.

6. Related to the point above it would be useful to discuss in some detail both in the intro (e.g. in lines 66-68, lines 96-98) and discussion (see below) how does the current work contrasts with recent computational models of cortico-cerebellar interactions (Pemberton et al. Nature Comms 2024, but also Boven et al. Nature Comms 2023). Those models capture observations of cortico-cerebellar interactions, while making several predictions. These models can be interpreted as a forward/predictive model and explains many observations, including motor preparation. Note that the Pemberton 2024 study is perhaps the most relevant as its closer to your model. For example, when you write "As the feedforward model requires feedback during the actual execution of the movement to optimize its future predictions.." in the discussion, it would be relevant to discuss this model. Moreover, "to the cerebral cortex to improve subsequent motor commands" is very consistent with

the model proposed by Pemberton et al. In general, the discussion is quite short, so it would benefit of this and other additions. But also the introduction.

We understand and appreciate the comments and suggestions for deepening and widening the Introduction and Discussion. Given the word limitations of Nature Communications we decided to focus on expanding the Discussion. For example, we now explicitly discuss the outcomes of the RNN-based models as highlighted by the reviewer.

Minor concerns:

1. "The argument of the transfer function is the incoming current for each population, and it contains a background term, a local term, and a long-range term." Could the authors clarify why this specific function was chosen instead of more commonly used activation functions such as sigmoid, ReLU, or tanh?

The transfer function used is common for neurobiologically-oriented models (e.g., Abbott and Chance, Prog Brain Res 2005). It offers a better approximation to the input-output function of spiking neurons than the ones given by sigmoid, ReLU, tanh or other functions. We have now indicated our motivation in the text.

2. What is the rationale for selecting a 20:80 proportion for LFP computation?

This choice stems from the fact that a typical electrode will tend to pick up electrical signals from deep pyramidal neurons more strongly than those from superficial ones, which are smaller and have shorter apical dendrites. A 20:80 ratio is a good approximation for our simplified model, which has worked well in the past (e.g., Mejias et al. 2016; Lindeman et al. 2021). We now explain this in the text.

3. The input to the model and its role in generating movement are not entirely clear. Does the model produce movement explicitly, or does it merely simulate neural activity related to movement?

As a network of neural-mass units, our model focuses on simulating firing rates across large-scale networks, rather than explicitly simulating microscopic neural activity and decode movement from it. We have clarified the nature and properties of the model in the text.

4. Could the authors elaborate on why laminar layer modules are critical for the movement control model being explored? How do they contribute to the conceptual framework?

The presence of laminar modules, even in the simplified form used here, constitute an essential ingredient of our model due to the fact that cortico-cortical interactions between M1 and S1 are strongly layer-dependent during whisking (Lindeman et al. 2021), and ignoring this would make replication of the results shown in Figure 2 not viable. We have now briefly mentioned our motivation in the text.

5. It would be helpful to mention the optogenetic stimulation periods in the results section where relevant.

The stimulation periods are now mentioned.

6. All t-tests should be reported with degrees of freedom, t-values, and effect sizes.

We added a Table with all the requested statistical values.

7. Significance markers in figures should indicate p-value levels explicitly.

We now indicate with the appropriate markers the p-values level explicitly.

8. In Figure 1c-e, is the data from a single animal, while Figure 1f represents the group-level data? Please specify this in the text.

This is now specified.

9. Figure 2a appears redundant.

We frankly agree; we have replaced the exemplary whisker raw trace that was identical to the one in Figure 1b. The new whisker trace shows that during the movement epoch, both retraction and protraction are possible. This also addresses major concern 2 of this Reviewer.

10. In Figure 2b-c, what exactly does the C-C coefficient represent? Is it the average cross-correlation in a 200 ms window across all pairs, computed per animal? Or is it the peak value?

It is the average peak value per animal. We extracted the peak value of the cross-correlogram per pair of signals and then calculated the average per condition. Now we replaced the y label in the figure and clarified it in the legend.

11. For Figure 1, how many trials contribute to each cross-correlation estimate per animal? Additionally, what is the standard deviation?

These values are now added to the new Methods section and to Figure S2.

Reviewer #2 (Remarks to the Author):

In this manuscript, Romano et al. studied the dynamic information flow between M1/S1 and the cerebellar cortex right before and after spontaneous whisker movements. Paired LFP recordings, optogenetic manipulations, computational modelling and spike recording techniques were used in order to support their conclusion. I have some concerns about the methodology, some of the results and interpretation, which I listed below.

Major concerns

1. Analysis of paired LFP recordings can suggest information flow between brain regions. However, I think relying mainly on LFP analysis is not enough. LFP signals integrate neural activities across a broad spatial scale and may introduce confounds when attempting to infer communications between brain regions. The authors should provide additional validation, possibly through spike analysis (mutual information, entropy...) and Granger causality.

We appreciate this constructive suggestion. We did more experiments and we added more analyses of more spike data in the new Supplementary Figure S4. We first calculated the level of mutual information to validate the LFP signal with the spikes presented in Supplementary Figure 1a,b and then we show mutual information between cerebral and cerebellar LFP signals compared to the spikes of single units. This new analysis suggests that using LFP might be more informative than neuronal spikes. In addition, we calculated the Granger causality between cortical and cerebellar spikes. The results show that the main direction of the Granger causality between M1 and PC is different during the Pre-movement compared to the Movement epoch. The results confirm that there is greater Granger causality from M1 to PC during the Pre-movement and from the PC to M1 during Movement epochs. The fact that this was not the case for the S1-PC spike is in line with the more consistent phase shift observed for Cb-M1 (Figure 1f, Figures S2-S4). Finally, we also performed a correlation matrix analysis based on trial-to-trial variability (Ten Brinke et al., 2015 Cell Reports). This analysis compares the spike rate of the two spike trains at several time delay intervals on a trial-by-trial base (Figure S4). This analysis showed that the spike rate of M1 predicts the subsequent spike rate of PC firing best during the Pre-movement stage and vice versa during the Movement stage. Taken together, these new data and analyses bolster our original interpretations.

Waveform of LFP signals depends on recording locations. While it's nice to see results from 16 channels, it's also obvious that they are quite different (e.g., Figs 1a, 6j). Instead of simply averaging across channels, the authors may want to have more analysis on individual channels. For example, signals from the 16 linear channels in M1 and S1 may be biased towards different layers. Since the information flow among different cortical layers have been previously well characterized, this analysis may be used to validate the methodology (i.e. determine information flow using paired LFP recordings).

We also fully agree with this point. To validate our main metric (i.e., phase relations or time displacement of the cross-correlograms) we did more analyses on individual channels as suggested. We compared the described input flow among different layers of S1 (Petreanu et al., 2009) and M1 (Hooks et al., 2011) with our data. We used current source density (CSD) analysis, which allows us to trace information flow across cortical layers, and we show that, in line with previous data, information flows from superficial to deeper layers (new Figure S3d). This phenomenon is captured by our metric (i.e., time displacement of the cross-correlograms), which shows consistent phase relations from

superficial to deep channels. Our metric also describes the information flow from M1 to S1 as described by Ahrens and Kleinfeld (2004; J. Neurophysiol.). The phase relation between all channels of S1 and M1 is such that superficial layers of M1 precede all channels of S1, as expected for information flowing from M1 to S1. From another project, we have data showing that our metrics capture also information flowing from S1 to M1 during whisker sensory stimulation. Even though we plan to publish these latter data and analyses in another publication with a different focus, they are still in line with the interpretations of the current manuscript.

We would like to clarify that we analysed each pair of channels individually, correlating all possible 384 combinations (16 M1 x 12 cerebellar and 16 S1 x 12 cerebellar) and that the main result was consistent across channels averaging the values of phase relation of the significant correlating pairs. We did average them, because, in a preliminary analysis, the cerebellar signal had a similar phase relation with the superficial and deep cortical channels. Now we also show the LPF signal dependency on the cortical location in the new Figures S2 and S3. Because the result of our new analysis on individual pairs of LFP signals is in line with existing literature, we believe that our methodology (i.e., analysis of the phase relation calculated from the cross-correlation) is a valid metric for assessing information flow within and between brain areas. We now better explain these points in the main text.

2. Some of the results may not support the final conclusion. i). A major conclusion of the manuscript is that during movement, LFP signals in Cb precede those in M1 (fig. 1f). However, in fig. 4f (top right, orange color), all results exhibit the opposite. ii). Statistics in fig. 2f and 2g may need double check. $P = 0.0496$ seems very weak. Is 3/55 and 7/62 enough to suggest stronger Purkinje cell activity correlation with S1? Are the layer distribution of these units in S1 and M1 comparable? iii). In line 270, it's stated "the polarity of the effect following this inhibition was opposite to that following excitation in both M1 and S1". However, in fig. 6j, results from channels 1-5 don't support this claim. This point may be worth discussion.

The Reviewer raises valid questions (i, ii and iii) with regard to three quantitative findings; we will address them consecutively.

*Ad i). We thank the Reviewer for their sharp eye. We double-checked the raw data and indeed found a mistake, for which we apologize; in several cases the inhibition was used as control and vice versa. After fixing this mistake, which occurred only in this panel, the results are more in line with the others as well as the general conclusion. Indeed, our conclusion that LFP signals in Cb preceded those of M1 during movement is now consistent in 12 out of 12 mice in the absence of stimulation (Figure 1f) and in 9 out of 9 mice during manipulation of the ascending pathway (Figure 5f top right). In addition, we also did the same experiments while stimulating other brain regions for control (i.e., stimulating other tracts than the descending route); during movement the LFP signals of the cerebellum in these controls also preceded those of cerebral cortex (see **Figure** below, for M1 and S1 $p = 0.004$ and $p = 0.008$, respectively; paired TTEST). Together, our data suggest that the repeated manipulation of the cortico-ponto-cerebellar pathway during the Pre-movement stage might have affected also the subsequent cerebrocerebellar signalling during the Movement stage. We now mention this possibility in the Discussion, while also highlighting that our data are in line with the directed coherence from the cerebellum to the prefrontal cortex during locomotion (Wang et al., 2023; Nat. Neurosc.).*

Ad ii). For the statistics in Fig. 2f and 2g we had a relatively low p-value, because it did not account for the fact that there were more significant pairs for S1 than M1. We now added a new panel in Figure 2, in which we consider both the number of significant pairs and the strength of the correlations. When we compare the amplitude of the peaks of all cross correlograms, instead of just the selected significant ones, the p-value = 0.003. In this way, we account for the fact that S1 has more pairs with strong Purkinje cell correlations and consider also the difference in the other pairs that did not cross our conservative threshold of significance. Please note though that our conclusion on the “stronger Purkinje cell activity correlation with S1” was also based on the LFP data of the 12 mice showing a stronger effect at all the cerebellar locations combined. There the p-values are 0.027 for the Pre-movement and 0.019 for the Movement epoch. We now show in the new panel of Figure 2 that the mutual information between spikes of Purkinje cells and S1 neurons is much higher than that between Purkinje cells and M1 neurons ($p = 0.0009$). Moreover, as suggested by the Reviewer, we have now also examined the layer distribution of these units in S1 and M1, and we report the comparison of the depth of the channels from which these neurons were recorded. Finally, in the Methods section we now mention that the depths of the recordings were comparable ($p = 0.684$).

Ad iii). We appreciate this point; we adjusted this claim and address it in the Discussion.

3. The rationale behind the computational modelling is not very clear. The authors should explain how this modelling advances our understanding of the system beyond what is achieved through experimental observations. From reading the current manuscript, I was left unclear of what extra insight the modelling provides.

This important question was also raised by the other Reviewer. Please see our answer to his/her point 5 above. In the revised version, we have adjusted the main text to better highlight our motivation for including the model, the insights it provided, and the choice of our modelling approach. We hope these changes make the value of our modeling efforts more transparent to the reader.

4. Looking into details of the modelling part, I have more questions. i). Can authors provide more details about the “preparatory ramping signals”? It looks to me that this setting is the major factor that results in different lagging values between pre-movement and movement. ii). If this is true, the authors may want to test different ramping parameters and check their effects on simulation results. iii). “an OU process arriving at deep layers of M1”. Does this setting actually cause the higher cerebellar correlation with S1? After all, $\sigma x = 100$ will make it VERY noisy. iiiii). Many of the parameters were manually determined. What’s the rationale behind these parameter settings? iiiiii). The authors later uploaded original codes for their modelling, which I really appreciate. In the codes, authors separate “Pre-movement” and “Movement” into different “conditions”. However, actual movement should be a continuous mixture of “Pre-movement” and “Movement”. I think this may be the place where computational modelling can kick in and provide insights.

The Reviewer is bringing up several interesting and important points.

(i) The preparatory ramping signals are modelled here as a noisy current to deep layers in M1; more specifically, they are modelled as an OU noise with a time constant $\tau = 200$ ms and $\sigma = 100$. The presence or absence of this signal is how the distinction between Pre-movement and Movement is introduced in the model; therefore, this signal is entirely responsible for the comparisons shown in Fig. 3b-d.

(ii) We have explored the effect of these parameters, and the results are consistent as long as the time constant for the preparatory signal is long enough (and σ is also large enough, see also our next point). The reason is that this preparatory signal is tasked with introducing slow, naturalistic fluctuations in the M1 firing rate, which can then drive activity along the descending pathway. In this sense, an OU noise signal with $\tau = 300$ ms would work as well, but one with $\tau = 1$ ms will be too fast to modulate M1 firing rates.

(iii) As mentioned above, this setting is responsible of embedding M1 with naturalistic slow variations in its otherwise constant firing rate level. The high cerebellar correlation with S1 is more related with the common sensory afferents, modeled here as "a shared OU process arriving at both PCs and S1", which is motivated by neuroanatomical evidence of sensory pathways targeting both PCs and S1, but not M1.

Regarding the value of $\sigma_x = 100$, it is useful to note that the timescale of the OU noise also affects its variance: an OU with $\sigma = 100$ and $\tau = 1$ ms would indeed be very noisy, but another with $\sigma = 100$ and $\tau = 300$ ms would have much less variance (about 20 times lower). Importantly, given the long time-constant, such an OU process will not be perceived by the system as 'fast noise', but rather as a slow driver modifying the firing rates.

(iv) The parameters were manually determined to set the model in a biologically reasonable working point: for example, parameters for cortical circuits were estimated from neuroanatomical and electrophysiological data, so as to ensure plausible cortical dynamics (Mejias et al., 2016). Parameters for thalamic and cerebellar pathways were chosen to ensure that changes in firing rates would trigger large enough changes in downstream areas, and within that range, to fit neuro-anatomical estimations and previous experimental recordings (Lindeman et al., 2021). Parameters for neural populations, such as time-constants and firing threshold values, were chosen from physiologically realistic ranges or from values that would provide realistic ranges. These parameter values are only rough estimations, because current datasets do not provide reliable values for all these parameters in the context of a large-scale cerebello-thalamo-cortical network.

(v) This distinction was made to better align with the experimental section of the paper, in which Pre-movement and Movement constitute two well defined categories to drive comparisons. While it is true that the real process would be an interesting continuous transition between the Pre-movement and Movement epoch, we think that the model unfortunately does not provide any compelling insight, other than that related to the overall differences beyond the transition period. Given that the preparatory signal is introduced in the model as an OU noise, there would be small time intervals in which the OU will slowly decrease until almost becoming zero (grossly equating the 'Movement' phase), while shortly before, it was at very high levels, marking a distinctive 'Premovement' phase. The natural conclusion would be that the Pre-movement activity is more dynamical than the Movement condition, but this has however more to do with our simplified modeling of the Premovement condition than with a solid prediction based on existing data.

To improve our exposition of all the points highlighted above, we have expanded the explanation of the OU signals in the main text, and we have expanded our rationale for choosing parameter values.

5. GtACR is known to induce antidromic spikes (PMID: 30297821). Considering the local connections in PN and TH, the authors may want to verify that light stimulation does not cause activation.

We thank the Reviewer for raising this important question. We would like to point out that we used the AAV1-hSyn1-SIO-stGtACR2-FusionRed (Addgene). Because we were aware of the possibility of generating antidromic spikes, we used the stGtACR2 (reported in the methods as AAV1-hSyn1-SIO-stGtACR2-FusionRed), instead of the classical GtACR2, and we stimulated somatically, rather than the distal axons. In the suggested publication (Mahn et al., 2018; Nat Commun.), it is shown that this approach reduces drastically antidromic spiking in vivo. We previously showed no antidromic spiking using this virus (Wang et al., 2023; Nat Neurosci.), but to make sure that this is the case also in the brain areas specifically targeted in this study, we tested whether spike responses were induced in neurons of M1 and S1 when stGtACR2 was expressed in the thalamocortical pathway and in Purkinje cells when stGtACR2 was expressed in ponto-cerebellar neurons. We now show that activation of stGtACR2 does not trigger spike activity in the new panels of Figure S6 (e-g) and we clarified that we used stGtACR2 and not GtACR2.

6. The authors suggest that the phase difference is consistent with the cerebellum receiving a copy of the cerebral commands during movement preparation, and the cerebellum provides feedback to the cortex during movement execution. While this is an interesting hypothesis, the forward model also suggests that the cerebral cortex need to send continuous motor command copies to the cerebellum during motor execution, in order to generate predicted sensory feedback for fast real-time motor adjustments. The representative results in fig. 1c on the right did exhibit double peaks both before and after movement onset. Is this pattern consistent?

In fact, it's also plausible that during the motor preparation, the cerebellum provides feedback to the cortex. Theoretically this can help with movement planning. I think it's OK to mention these and other hypothesis in the discussion part of the manuscript. However, if claiming these points in the other parts of the manuscript, more work will be needed.

To assess whether the presence of double peaks in the cross-correlograms was a consistent pattern we quantified their occurrence and now we report it for each condition (Figure S2 panel f). Because only 15.7% of the cross-correlograms had double peaks, we concluded that this is not the main pattern. However, we now acknowledge the possibility that the cerebellar activity could help with movement planning in the Discussion, because, even if this is not the main pattern under our experimental conditions, we cannot exclude that this could happen in different experimental conditions, such as goal-oriented behavior (see also one of our previous findings exploiting preparation of directional tongue movements in Gao et al., 2018; Nature).

Minor concerns:

1. What's the correlation coefficient for data in Figs 4 and 5?

It is a parameter returned by the MATLAB function xcorr to calculate the cross-correlogram. Because we were interested mostly in the timing of the maximal peak, we normalized the cross correlogram of the different simulations so that they appear with similar amplitude. Now we added a sentence in the legend to further clarify this.

2. Spontaneous whisker movement is likely not a goal-directed voluntary movement. Can the observations here be generalized to other types of movement? Some discussions of this topic may be helpful.

This is important indeed; we now discuss this possibility.

3. The volume of viral injections seem to be all wrong in the methods part. It's hard to imagine that 30-40 ul of virus can be infused to a single brain region...

We thank the Reviewer for catching this mistake; we fixed it.

4. The manuscript lacks a detailed description of how the two recording systems were synchronized. Precise timing is critical in this study.

We now clarified in the Methods that the signals from all electrodes and probes were acquired using a single Tucker-Davis Technologies recording system, which ensures synchronization at a high resolution.

5. In figure 1, it should be fig. 1f instead of 1F.

Done!

6. What's the titer of AAVs used in this study?

We now report the titer in the Methods.

Reviewer #3 (Remarks to the Author):

Reviewer #1 (Remarks to the Author):

We thank the reviewers for addressing our comments in detail. We are happy with the revised manuscript.

Reviewer #2 (Remarks to the Author):

The revised version addressed most of my concerns. Here I listed a few points which hopefully are helpful for the authors.

Firstly, the “control” group in Figure 4f and 5f need to be defined clearly. I initially wrote a long paragraph explaining why data in Figure 4f still doesn’t fit with the conclusion that Cb signals precede M1 during movement. Later I deleted it, because I realized that the “control” condition in this study is not fluorescent protein expression plus light on, but stGtACR2 expression plus light off (is that right?). And now the response in rebuttal letter all makes sense to me. Please clearly define the “control” condition in optogenetics experiments, so that some readers don’t need to experience the confusion I do. Although I still think fluorescent protein control is better for optogenetics experiments (as the authors suggested, repeated manipulation have after-effects), this is more of a suggestion for future experiments, rather than a suggestion for revision, since it doesn’t affect the conclusion of this study.

Yes, the Reviewer is correct! We now define our “control” condition in the Results, Methods and Legends. We thank the Reviewer for bringing this up.

Secondly, the authors need to double check their statistical methods. Currently all statistical analysis is based on t-test. However, some data obviously don’t follow a normal distribution (e.g., Figure 2h).

We have checked for normality and applied the non-parametric test where it was needed. We also thank the Reviewer for this suggestion.

Lastly, the authors need to be more careful with manuscript writing. I listed some errors that I noticed during reading. But please proofread the manuscript on your side.

- 1). Figure legend for Figure 2h is completely missing; *We added it.*
- 2). Line 30, “align actual with”. This expression seems not right; *We changed this.*
- 3). Line 166, it should be “Figure 2f” instead of “Figure 2e”; *Done!*
- 4). Line 430, it should be “fine-tuning” instead of “finetuning”; *Done!*
- 5). Line 483, “similar to 13”. This citation format seems not right; *We changed this.*
- 6). Line 489, “see 23”. Same citation format issue; *We also changed this.*
- 7). Line 507, it should be “unit” instead of “unite”; *Done!*
- 8). In figures, “GtACR2” should be replaced with “stGtACR2”. *Done!*

We thank the Reviewer for carefully checking our manuscript and suggesting how to improve the manuscript. We have adhered to all suggestions, including that of proofreading.